METHODS

# Learning spatio-temporal patterns with Neural Cellular Automata

**Alex D. Richardson**[1,2]*, **Tibor Antal**[2], **Richard A. Blythe**[1], **Linus J. Schumacher**[2,3]

**1** School of Physics and Astronomy, University of Edinburgh, Edinburgh, United Kingdom, **2** School of Mathematics and Maxwell Institute for Mathematical Sciences, University of Edinburgh, Edinburgh, United Kingdom, **3** Institute of regeneration and repair, University of Edinburgh, Edinburgh, United Kingdom

* alexander.richardson@gmail.com

**Data Availability Statement:** All code and data is available at: https://github.com/AlexDR1998/NCA.

**Funding:** ADR received funding through a PhD stipend paid by the EPSRC Centre for Doctoral Training in Mathematical Modelling, Analysis and

## Abstract

Neural Cellular Automata (NCA) are a powerful combination of machine learning and mechanistic modelling. We train NCA to learn complex dynamics from time series of images and Partial Differential Equation (PDE) trajectories. Our method is designed to identify underlying local rules that govern large scale dynamic emergent behaviours. Previous work on NCA focuses on learning rules that give stationary emergent structures. We extend NCA to capture both transient and stable structures within the same system, as well as learning rules that capture the dynamics of Turing pattern formation in nonlinear PDEs. We demonstrate that NCA can generalise very well beyond their PDE training data, we show how to constrain NCA to respect given symmetries, and we explore the effects of associated hyperparameters on model performance and stability. Being able to learn arbitrary dynamics gives NCA great potential as a data driven modelling framework, especially for modelling biological pattern formation.

## Author summary

Pattern formation is ubiquitous in biological systems, across many length and time scales—from vegetation stripes in deserts, to the spots of a leopard's skin, and the fine detail of stem cell differentiation in an embryo. While many simple rules that create complex patterns are known, reverse engineering the mechanisms responsible from an observed pattern has generally remained a difficult problem. In this work we build on the connections between machine learning, cellular automata, and partial differential equations, to create models that can learn the underlying mechanisms that yield any desired emergent pattern. To do this we consider Neural Cellular Automata (a mix of neural networks and cellular automata). We describe how they are built and trained, and we show that they are capable of reproducing a wide range of complex emergent behaviours. Previous work focuses on learning stationary patterns, but we focus on patterns that evolve in time. We show these models can learn classic Turing patterns (biochemical models of spot and stripe formation), as well as learning to morph through an arbitrary sequence of images. We believe this hybrid between machine learning and

Computation (MAC-MIGS) (grant number: EP/S023291/1, url: https://www.ukri.org/councils/epsrc/). The funder had no role in study design, data collection and analysis, decision to publish, or preparation of the manuscript.

**Competing interests:** The authors have declared that no competing interests exist.

mechanistic modelling has great potential for modelling biological growth, regeneration, and pattern formation.

This is a *PLOS Computational Biology* Methods paper.

## 1 Introduction

Many complex natural phenomena—such as organ growth, the structure of materials or the patterns of neural activity in our brains—are emergent [1]. These are typically characterised by many simple interacting components that collectively exhibit behaviour that is far richer than that of the individual parts, and cannot easily be predicted from them. Emergence is especially prevalent in complex systems of biological nature across a wide range of scales—from gene expression dictating cell fates, interacting cells forming structures during morphogenesis, synaptic connections in the brain, or the interactions of organisms in ecology.

Cellular Automata (CA) provide simple models of spatio-temporal emergent behaviour, where a discrete lattice of 'cells' are equipped with an internal state and a rule that updates each cell state depending on itself and its local neighbours. The classic *Game of Life* [2] is a famous example, where cell states and the update rule utilise simple Boolean logic, but the emergent complexity has fascinated and inspired much research [3, 4]. CA are a natural modelling framework of a wide range of biological processes such as: skin patterning [5, 6], limb polydactyly [7], chimerism [8], cancer [9] and landscape ecology [10]. In these cases the CA rules are constructed with expert knowledge of likely mechanisms, however in general the space of possible CA rules is vast, and there is a non-uniqueness by which several rules can result in qualitatively similar emergent behaviours. As such the inverse problem of inferring mechanistic interactions (CA rules) that might generate a given observed emergent behaviour is much more challenging than the forward problem. Establishing the emergent consequences of a known set of mechanistic interactions between components is conceptually straightforward—one sets them up in a computational model or *in vivo* and then observes the collective behaviour that emerges. In the case of CA, once a rule is defined, any initial condition can trivially be propagated forward to obtain the emergent behaviour.

In this work, we establish and extend the utility of Neural Cellular Automata (NCA, [11])—a special case of CA where each cell state is a real vector, and the update rule is determined by a neural network. The update rule is parameterised by the neural network by minimising a cost function that measures how similar the trajectory generated by iteratively applying the update rule is to training data. Since the gradient of the loss function can be evaluated straightforwardly, gradient-based optimisation allows for efficiently learning (non-unique) local update rules to match desired global behaviour, in the usual manner of machine learning [12]. This allows us to tackle the inverse problem of inferring local mechanistic rules given observed emergent behaviour.

We investigate the potential for NCA to be applied as a data-driven alternative, where the underlying mechanisms are not assumed to be known. This could include biological systems, which are inherently complex and may feature interactions that are not directly measured by a given experimental procedure. We do this by exploring the behaviour of NCAs on two idealised systems. First, we train the NCA on Turing patterns generated by the solutions of certain

partial differential equations (PDEs). We show that the underlying dynamics are well represented, for example, by testing on initial conditions that were not part of the training set. Second, to understand the generality of the method, we build on previously observed behaviour of NCA on the artificial problem of morphing from one image to another [11]. Here any underlying dynamics is *a priori* unknown and likely to be highly complex, as opposed to the PDE learning case where we know and understand the PDE. Nonetheless we show that such dynamics are learnable and are robust to perturbations. In addition, we achieve this with the minimal neural network complexity of NCA [13], in line with the principles of Explainable Artificial Intelligence [14, 15].

In Section 2 we set out the structure of the NCA, and show that it can be viewed as a machine learning approach to approximating the finite difference discretisation of an unknown PDE. There is already a fairly strong link between cellular automata and reaction diffusion equations [5, 6], in that both are used to model similar systems, and a suitably chosen CA rule will correspond to the finite difference approximation of any PDE. In Section 3 we present our results. We first (Section 3.1) benchmark the NCA by assessing its ability to capture the types of Turing patterns [16] that emerge from the Gray-Scott [17] reaction-diffusion equations. These equations describe the population of two chemical species with a nonlinear interaction between them, and is capable of generating a variety of patterns. In Section 3.2, we show that the same basic model and training techniques can also be applied to an image morphing task. Thus we conclude that NCA are capable of constructing microscopic dynamical rules that lead to a wide range of prescribed emergent behaviour. In Section 3.2.2 we further explore constraining NCA to respect basic symmetry requirements placed upon them, and investigate the robustness of trained NCA to initial condition perturbations. We discuss implications for further development of the method Section 4.

## 2 Model and methods

We now define the NCA model, and discuss the motivations behind design choices and hyperparameters. We further discuss the training methods, as it turns out that most training parameters can be kept constant between tasks. The main exception to this is how frequently to sample in time: PDEs have a clear notion of time, whereas image morphing does not. All the models and software developed here have been implemented within the Tensorflow [18] framework https://github.com/AlexDR1998/NCA which permits efficient GPU parallelisation.

### 2.1 Model details and parameters

Neural Cellular Automata (NCA) are a class of cellular automata defined on a lattice of real vectors, with the local update rule encoded in a neural network. As with all neural network models, there is freedom to choose the network structure and how the input data are preprocessed for training and testing. We refer to the set of choices that are not learned via the training data as *hyperparameters*. Fig 1 shows a schematic of a single NCA update, which maps the state of the system at time step $n$ to a corresponding state at time $n + 1$, where $n = 0, 1, . . .,$. This update comprises a sequence of stages—depicted counterclockwise starting top left in the figure—which we now describe in turn.

**A: System state.**   We take the state of the system to be described through a vector of $C$ real numbers at each point on an $S \times S$ lattice, as shown in Fig 1A. For example, each of the $C$ values could represent the concentration of a chemical or biological species at a given point in space. These observable channels are shown with coloured shading throughout Fig 1. These can be augmented with *hidden channels* (transparent in the figure), the state of which can influence the dynamics within the *observable channels*. In a biological context, these hidden channels

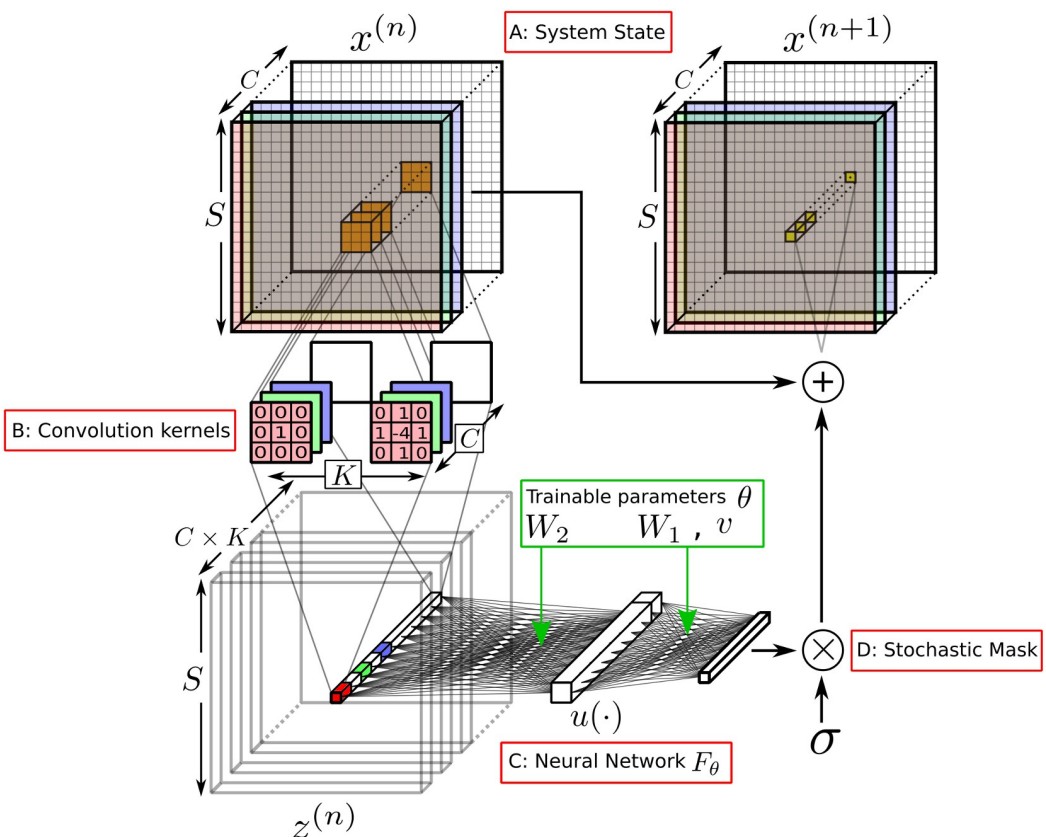

**Fig 1. Schematic of an update step of the NCA.** For each $C$ channel pixel in the $S \times S$ lattice $x^{(n)}$ at step $n$, a perception vector $z^{(n)}$ is constructed to encode local information via convolution with hard-coded kernels $K$. This perception vector is fed through a dense neural network $F_\theta$ with trainable weights $W_1$, $W_2$, and biases $v$. The nonlinear activation function $u(\cdot)$ is applied on the single hidden layer of the network. The output of this network yields the incremental update to that pixel, which is applied in parallel to all pixels with the stochastic mask $\sigma$ to determine the lattice state $x^{(n+1)}$ at step $n + 1$.

could represent concentration profiles of proteins or chemicals that are not measured in a particular experimental setting, but can be inferred by the machine learning algorithm. The number of hidden channels is a hyperparameter of the model. The number of hidden and observable channels sum to $C$. Mathematically, we denote the state of the system at timestep $n$ as the vector $x^{(n)} \in \mathcal{X} = \mathbb{I}^{S \times S \times C}$, where $\mathbb{I} \in [a, b]$ is some interval of real numbers. The elements of this vector are $x_{ijc}^{(n)} \in \mathbb{I}$, which $i \in [1, S]$, $j \in [1, S]$ and $c \in [1, C]$ denote the $x$-coordinate, $y$-coordinate and channel number, respectively. We emphasise that during training (Section 2.2), only the observable channels are compared to data.

**B: Convolution kernels.**   The first stage of the update is to apply *convolution kernels* to the spatial data in each channel. These kernels are chosen as discretised differential operators, such as gradients and Laplacians. We denote the set of convolution kernels $g^k$, labelled with the index $k = 1, \ldots, K$. Each kernel $g^k$ is a square matrix, in our case ($3 \times 3$). See Fig 1B. We use Sobel filters as gradient approximations. This generates the expanded *perception vector* $z^{(n)}$ whose elements are given by:

$$z_{ijck}^{(n)} = \sum_{\Delta i, \Delta j \in [-1, 0, 1]} g_{\Delta i, \Delta j}^k x_{i+\Delta i, j+\Delta j, c}^{(n)} \equiv g * x^{(n)} \tag{1}$$

Crucially the kernels are applied in parallel on all channels $C$ independently: that is, all kernels are applied *depthwise*. The idea of decomposing an arbitrary convolution to separate depthwise and channel-mixing convolutions [19] was inspired by the deep link between Convolutional Neural Networks (CNNs) and Cellular Automata [11, 20]. In particular, this facilitates representing the NCA in standard Tensorflow code. In principle, kernels can be learnable rather than hard-coded; however this makes the trained models less interpretable and so we do not pursue this approach here. As the $3 \times 3$ convolution kernels only encode information about the Moore neighbourhood (i.e. adjacent and diagonal cells), we would never need any more than 9 kernels, as any more would be linearly dependant on those already present.

The purpose of applying the kernels is to make a clearer correspondence between NCAs and numerical methods for solving PDEs. Essentially, they provide the neural network with a basic set of differential operators to work with. The set of kernels to include is an important hyperparameter. For example, if one anticipates that the update rules should be invariant under a global rotation of the system—i.e., that the dynamics are isotropic—one can justify excluding gradient kernels and just using identity, Laplacian, and local averages. We already have translational invariance as a direct consequence of the NCA construction, but isotropic symmetry can only be realistically achieved by only using isotropic kernels [21] or data augmentation. We explore this further in Section 3.2.2. The explicit forms of the convolution kernels used in this work are

$$
\underbrace{\begin{bmatrix} 0 & 0 & 0 \\ 0 & 1 & 0 \\ 0 & 0 & 0 \end{bmatrix}}_{\text{identity}}, \quad
\underbrace{\frac{1}{9}\begin{bmatrix} 1 & 1 & 1 \\ 1 & 1 & 1 \\ 1 & 1 & 1 \end{bmatrix}}_{\text{average}}, \quad
\underbrace{\frac{1}{8}\begin{bmatrix} 1 & 2 & 1 \\ 0 & 0 & 0 \\ -1 & -2 & -1 \end{bmatrix}}_{\text{gradient}_x}, \quad
\underbrace{\frac{1}{8}\begin{bmatrix} 1 & 0 & -1 \\ 2 & 0 & -2 \\ 1 & 0 & -1 \end{bmatrix}}_{\text{gradient}_y}, \quad
\underbrace{\frac{1}{4}\begin{bmatrix} 1 & 2 & 1 \\ 2 & -12 & 2 \\ 1 & 2 & 1 \end{bmatrix}}_{\text{Laplacian}}.
\tag{2}
$$

**C: Neural network.**  The perception vector $z^{(n)}$ is then applied to the input layer of a neural network (see Fig 1C). The values on the output layer, $F(z^{(n)})$, form a vector of increments in the original state space $\mathcal{X}$. In a deterministic update, one would simply add $F(z^{(n)})$ to $x^{(n)}$ to obtain the updated state, $x^{(n+1)}$. Taking $F(z^{(n)})$ as an increment, rather than the new state vector $x^{(n+1)}$, implies that the NCA is a residual, rather than a naive, recurrent neural network (RCNN) [22].

In the present context, residual RCNNs have several benefits. Firstly a residual RCNN is easier to compare to numerical discretisations of PDEs, aiding interpretability of our models. Secondly the residual RCNN minimises the problem of vanishing or exploding gradients that the naive RCNN would experience. In the naive approach, recurrent iterations of our neural network would lead to exponentially large or small gradient updates, leading to a failure to learn optimal model weights. A consequence of the vanishing gradients problem is that information from previous timesteps is quickly lost, so $x^{(m)}$ has little bearing on $x^{(n)}$ for $m \ll n$. In principle a naive RCNN can learn long term dependencies, but in practice this is very challenging. As such residual RCNNs are better suited to learning long term behaviours, as $x^{(n+1)}$ depends linearly on $x^{(n)}$, so information preservation is in some sense the default behaviour of our model. This behaviour is especially clear during training, in that initialising the residual RCNN to perform the identity mapping $x^{(n+1)} = x^{(n)}$ is straightforward: one simply arranges for $F(z^{(n)}) = 0$ by setting the final layer weights in the neural network to zero. Initialising the weights such that $F(z^{(n)}) = x^{(n)}$, which would be required in the naive case, is much harder. This 'do nothing' update is a better starting point than one that quickly forgets the initial state of the system. In the latter case, the NCA may resort to learning how to construct the desired

$x^{(n)}$ as a global attractor, irrespective of initial conditions, for example 'growing' $x^{(n)}$ from fixed boundary conditions. Preserving the initial system state allows the model to better learn dynamics particular to those initial conditions, whilst still allowing for boundary driven behaviour to be learned.

It remains to specify the neural network structure, that is, the number and size of any hidden layers, and how they are connected. Here we aim to keep the architecture as simple as possible, as a minimal yet sufficiently functional network architecture has several advantages. Training a small model is computationally cheaper, and smaller models are far more interpretable [15]. Specifically, we use just one hidden layer, as shown in Fig 1C. As noted previously, we do not mix spatial data in the neural network, only between channels and kernels. That is, the network shown in Fig 1C is replicated for each pixel $i, j$ in $z^{(n)}$, and takes as input the $K \times C$ elements of $z^{(n)}$ that correspond to a given pixel, and transforms to $C$ channel values, consistent with the original state vector.

The hyperparameters associated with this neural network structure are the choice of activation function and size of the hidden layer. We fix the hidden layer size to $H = 4C$ where $C$ is the number of channels. This way the network size scales with the number of channels, so we just explore them together as one hyperparameter. Denoting the hidden-layer activation function as $u$, we can specify the mapping $F$ through the elements of the output vector as

$$f_{ijc}^{(n)} = \sum_{h \in [1,H]} \left( W_1^{ch} u \left( \sum_{\substack{c' \in [1,C] \\ k \in [1,K]}} W_2^{c'kh} z_{ijc'k}^{(n)} \right) \right) + v^c \equiv F(z^{(n)}). \tag{3}$$

The weights $W_2$ mix information between the channels and kernels independently of the position $i, j$ to determine the activation of each hidden node $h$. The weights $W_1$ then combine the hidden nodes to construct the output value for each channel $c$, again independently of $i, j$. We emphasise that the same set of weights and biases is applied at every pixel, consistent with the separation of spatial and channel mixing between the two stages of the process.

**D: Stochastic mask.** The final step is to increment a random subset of state vector elements $x_{ijc}^{(n)}$ by applying a *mask* $\sigma^{(n)} = (\sigma_{ijc}^{(n)})$ (Fig 1D), where $\sigma_{ijc}^{(n)}$ are independent Bernoulli random variables with parameter $1 - p$. That is $\sigma_{ijc}^{(n)} = 0$ with probability $p$, which is related to the dropout rate in machine learning regularisation, and $\sigma_{ijc}^{(n)} = 1$ otherwise. The purpose of this mask is to break any global synchronisation between cells [11].

Given the above, the update specified by the NCA is

$$x^{(n+1)} = x^{(n)} + \sigma^{(n)} F(g * x^{(n)})) \equiv \Phi(x^{(n)}, \theta) \tag{4}$$

where we have introduced the mapping $\Phi(\cdot, \theta) : \mathcal{X} \to \mathcal{X}$ from one state vector to the next, where $\theta$ encodes the network parameters $(W_1, W_2, v)$. In terms of individual elements, this corresponds to

$$x_{ijc}^{(n+1)} = x_{ijc}^{(n)} + \sigma_{ijc}^{(n)} f_{ijc}^{(n)} \tag{5}$$

where the elements $f_{ijc}^{(n)}$ are given by Eq (3) above. See line 7 of Algorithm 1. Hence, given $x^{(0)}$, the NCA provides recursively the sequence $(x^{(n)})_{n = 0,1,2, \ldots, N}$.

**Algorithm 1** Pseudocode description of a single NCA update step. Here $x$ is an $S \times S$ lattice with $C$ channels. $g * x$ represent the convolutions described in Eq 1. $W_1$ and $W_2$ are the neural network weight matrices, with $v$ being a vector of biases, all of which are encoded in $\theta$. $u()$ is the activation function. Note that in practice the For loops in line 4 and convolutions in line 3

are efficiently parallelised in Tensorflow's GPU implementation. Random() samples a uniform $[0, 1)$ distribution.

```
1: function Φ(z, θ)
2:    W₁, W₂, v ← θ
3:     z ← g * x
4:    for all (i, j) ∈ [1, S]² do
5:      dx ← W₁u(W₂ • z[i, j]) + v
6:      if Random() ≥ p then
7:        x[i, j] ← x[i, j] + dx
8:      end if
9:    end for
10:    return x
11: end function
```

**NCA as a PDE discretisation.**   We have described how NCA relate to common neural network architectures, but as noted above we can also define them in terms of a discretised PDE. Consider discretising this PDE in time with Euler's method, and in space with finite differences:

$$\partial_t x = F(x, \nabla x, \nabla^2 x) \qquad \Rightarrow \qquad x^{(n+1)} = x^{(n)} + hF(g * x^{(n)}),$$

where we have a stepsize $h$ from the time discretisation which can be absorbed into the parameters of $F$ and the convolution kernels $g$, which include the identity and spatial discretised gradients and Laplacian of $x$. The discretisation takes the form of the NCA with $p = 0$ (no stochastic mask). Note that this PDE is very general, encompassing reaction-diffusion systems as well as fluid dynamics, depending on how $F$ combines spatial gradients.

**Batch parallelism.**   When implementing this model in Tensorflow, we make use of *batch parallelism*, where instead of training on one trajectory $(x^{(n)})_n$, we train simultaneously on a set (batch) of trajectories $(x^{(n, r)})_{n,r}$, where superscript $r = 1, 2, \ldots, R$ denotes the batch number. In effect this just adds an extra batch dimension to $x^{(n)}$, so $x_{ijc}^{(n)}, z_{ijdk}^{(n)}$ and $f_{ijc}^{(n)}$ become $x_{ijc}^{(n,r)}, z_{ijdk}^{(n,r)}$ and $f_{ijc}^{(n,r)}$ respectively. This is normally done to leverage low-level speed-up, as training the network on batches involves matrix-matrix rather than matrix-vector multiplications, which are well optimised on parallel architectures (GPUs). However in the case of NCA, batch parallelism enables far more diverse systems to be learned, for example learning several distinct trajectories by the same rule, or improving stability through data augmentation. It is also crucial for extending the existing NCA training algorithm [11] to learning longer sequences of data, as discussed in section 2.2.

To summarise, the NCA can be described in terms of neural network language as a residual (calculating increments to each pixel) Recurrent Convolutional Neural Network (RCNN) with per-pixel dropout (stochastic mask). It can also be defined as a discretisation of a PDE, which may help interpretability of the trained NCA. The hyperparameters are the number of hidden channels, the set of convolution kernels and activation functions on the hidden layer of the network. The effect of varying the hyperparameters is explored in Section 3.

## 2.2 Training techniques

The neural network architecture described above is very minimal in comparison to the state-of-the-art in the literature [23], featuring only a single hidden layer applied in parallel. By contrast, the training process is fairly complex. We set out key steps below, with corresponding pseudo-code set out as Algorithm 2. NCA trajectories can be considered as paths in $\mathcal{X}$ (Fig 2), where the training process constrains the NCA parameters such that the trajectories pass as close (defined by the loss function) as possible to the observed data points in $\mathcal{X}$. Projecting

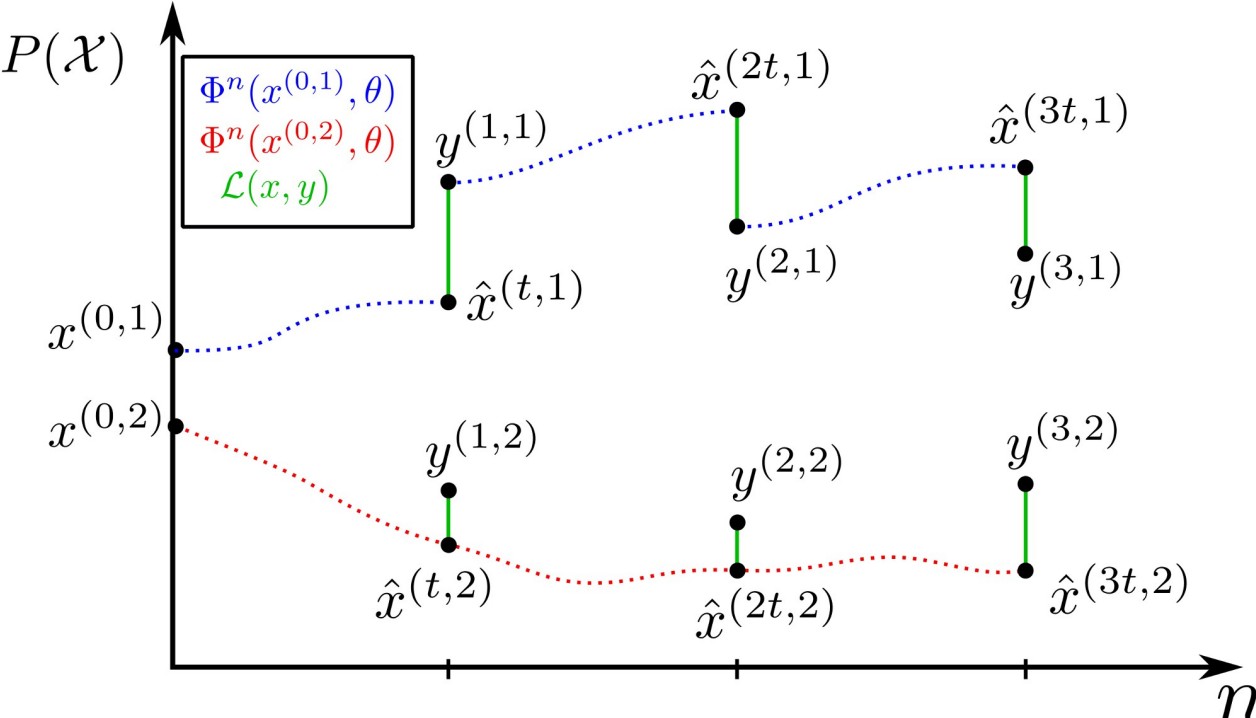

**Fig 2. 1D phase space representation of NCA trajectories, predictions $\hat{x}^{(mt,r)}$ and true states $y^{(m, r)}$.** Here $M = 3$, $R = 2$. The first batch $(x^{(\cdot,1)})$ is trained with re-initialised intermediate states, whereas the second batch $(x^{(\cdot,2)})$ is trained with propagated intermediate states.

these paths onto 1D helps visualise the training process, especially when training to multiple batches.

The technique of training NCA is based on backpropagation through time, a typical method for training RNNs [24]. We have established the batch of NCA trajectories $(x^{(n, r)})_{n=1,...,N}$, with $n$ and $r$ denoting time and batch respectively. Originally [11], training the NCA consisted of one set of initial states being mapped to one set of final states $x^{(0,r)} \rightarrow x^{(N, r)}$. We extend this to learn the set of transitions $x^{(n-1, r)} \rightarrow x^{(n, r)}$, where the batch parallelism allows us to train the NCA on each transition simultaneously. This allows training NCA to far more diverse and complex dynamics. For clarity we drop the explicit batch index (i.e. set $r = 1$) for now, the context where it matters is discussed later, but even so batch parallelism is still exploited for learning the different timesteps.

We have a trajectory of $M$ data points, indexed by the integer $m$: $(y^{(m)})_{m = 0,1, ..., M}$. Each $y^{(m)}$ is separated by $t$ NCA timesteps, that is $x^{(n)}$ corresponds to $y^{(m)}$ when $n = mt$ for $m = 0, 1, ..., M$ and $n = 0, 1, ..., N$, where $N = Mt$. Hence we need to compare predicted $\hat{x}^{(mt)}$ to $y^{(m)}$. We initialise $x^{(mt)} = y^{(m)}$ for $m = 0, ..., M - 1$, and propagate each state through $\Phi^t(\cdot, \theta) = \Phi \circ ... \circ \Phi(\cdot, \theta)$ ($t$ nested function compositions): $\hat{x}^{(mt)} = \Phi^t(y^{(m)}, \theta)$. To compute the loss, we compare $\hat{x}^{(mt)}$ to $y^{(m)}$ for $m = 1, ..., M$, averaging over different $m$ values: $\frac{1}{M}\sum_{m=1}^{M} \mathcal{L}(\hat{x}^{(mt)}, y^{(m)})$, where the loss function $\mathcal{L} : \mathcal{X}^2 \rightarrow \mathbb{R}$ is a meaningful measure of distance between any pair of $\hat{x}^{(i)}$ and $y^{(j)}$. Training is achieved by minimising the loss function, which requires partial gradients with respect to the trainable parameters $\theta$ to be evaluated. For the full gradient calculations, see S1 Appendix.

**Algorithm 2** Training an NCA $\Phi$ to $R$ data trajectories of length $M$. Split into $B$ mini-batches to reduce memory usage. Here $x^{(n, r)}$ and $y^{(n, r)}$ denote predicted state and data at step $n$ of trajectory $r$ respectively. $\hat{x}^{(n,r)}$ is a temporary variable to store new intermediate states at each training iteration. Lines 20 and 21 perform gradient normalisation followed by a parameter update handled by the Nadam algorithm (or any other optimiser of choice). The nested For Loops on lines 9,11 and 12 are easily parallelised. Typical choices of mini-batching are such that $10 < (M \times R)//B < 100$. $\Phi^t$ denotes $t$ iterations of $\Phi$. The choice of $t$ is an important hyperparameter, and relates to the temporal resolution of the data $x$. The model parameters $\theta$ are assumed to either be initialised appropriately, or already partially trained. RandomShuffle($A$, $B$) randomly shuffles $A$ and splits it into $B$ equal sized chunks. $\mathcal{L}(A, B) : \mathcal{X} \times \mathcal{X} \to \mathbb{R}$ computes the loss between states $A$ and $B$.

```
 1: function TRAIN(Φ, θ, y, t,B,EPOCHS)
 2:    x ← y
 3:    M ← y.shape[0]
 4:    R ← y.shape[1]
 5:    for i∈ EPOCHS do
 6:       Grad ← 0⃗
 7:       MS ← RandomShuffle([1,M],B)
 8:       RS ← RandomShuffle([1,R],B)
 9:       for b ∈ [1, B] do
10:          Loss ←0
11:          for m∈ MS[b] do
12:             for r∈ RS[b] do
13:                x̂⁽ᵐ,ʳ⁾ ← Φᵗ(x⁽ᵐ⁻¹,ʳ⁾,θ)
14:                Loss ← Loss + 𝓛(x̂⁽ᵐ,ʳ⁾,y⁽ᵐ,ʳ⁾)
15:             end for
16:          end for
17:          Loss ← 1/(M×R) Loss
18:          Grad ← Grad + ∂Loss/∂θ
19:       end for
20:       Grad ← Norm(Grad)
21:       Update(θ,Grad,i)
22:       for m ∈ [1, M] do
23:          for r ∈ [2, R] do
24:             x⁽ᵐ, ʳ⁾ ← x⁽ᵐ, ʳ⁾
25:          end for
26:       end for
27:    end for
28:    return Φ
29: end function
```

There are additional practical considerations for optimising this training method. After each iteration, we have the choice of keeping and further propagating the values $x^{(mt)} = \hat{x}^{((m-1)t)}$ for $m = 1\ldots M$, or re-initialising them: $x^{(mt)} = y^{(m)}$. Propagating them allows the NCA to better learn long term dynamics (particularly of the hidden channels) over many training iterations. However, we observe that re-initialising helps speed up the training process in the earlier steps. As both approaches have their advantages, we return to the batch parallelised case and do both. We regard re-initialising the states as a form of data augmentation (Fig 2), so in practice we only re-initialise one batch: $x^{(mt,1)} = y^{(m,1)}$. This choice of only re-initialising one batch performs well, but is arbitrary and could be further tuned for specific problems. The NCA is initialised with random hidden layer weights ($W_2$), and zero final layer weights ($W_1$) and bias ($v$).

When implementing the training procedure, as described in Algorithm 2, the additional subtlety of mini-batching is required [25, 26]. Rather than computing the gradient for transitions $x^{(m-1, r)} \to x^{(m, r)}$ for all $m \in [1, M]$, $r \in [1, R]$, we randomly split $[1, M] \times [1, R]$ into $B$

*mini-batches*, and separately compute the loss gradient for each mini-batch. After iterating through each mini-batch, the gradients are averaged and applied once to the model parameters. The need for mini-batching is due to memory constraints: if a large enough number of batches $R$ or timesteps $M$ is used, computing the gradient over the full set of transitions is unfeasible. In Algorithm 2 the memory cost of calculating the gradient (line 18) scales like $M \times R \times S^4 \times C^2 \times t$ (where $|\mathcal{X}| = S^2C$). By contrast the memory cost of storing and adding to the calculated gradient over each mini-batch is fixed as $\|\theta\|$ (i.e. does not scale with $B$), which is minimal given the small size of the network. $S$ and $C$ are fixed by the spatial (and channel) resolution of the data, but mini-batching reduces the memory burden of $M$ and $R$. For the full calculation see S1 Appendix. In the case of $B = 1$, the mini-batching reduces such that the For loops at lines 9, 11 and 12 collapse into simpler loops over $[1, M]$ and $[1, R]$. When training on image morphing [11], the size of data does not require mini-batching as $M$ and $R$ are small. To accurately capture PDE dynamics, much larger $M$ is used, and as such mini-batching is required. We do not explore spatial mini-batching, where random subsets of pixels are tracked for gradients, as this makes efficient parallelisation more challenging, however it could be very useful to further explore as it could enable training of NCA to higher resolution data.

**2.2.1 Loss functions.**   We now turn to the question of how to define the distance between a NCA trajectory and target data. This problem is split naturally into two parts: first, how to find the difference between corresponding (time- and batch-labelled) points in $\mathcal{X}$, and then how to combine all these difference measures to the distance between two sets of points in $\mathcal{X}$. For the latter choice, we adopt an arithmetic mean over the time points and batches considered (as shown in lines 14 and 17 of Algorithm 2), although these could be weighted (e.g., the states at a specific time or batch being most important).

We compared several loss functions for corresponding points in $\mathcal{X}$, but we found that the standard Euclidean norm $\mathcal{L}(x, y) = \left(\sum_{i,j,c}(x_{ijc} - y_{ijc})^2\right)^{\frac{1}{2}}$ worked best in all contexts. Probability mass based losses (Hellinger [27] and Bhattacharyya [28]) and distance between spatial Fourier transforms of points in $\mathcal{X}$ all work well in the PDE modelling case, but perform poorly on image morphing. Various approximations to the Wasserstein distance [29] performed poorly in both contexts but still remain promising given their success in texture synthesis [30–32]. As such, for the following results we stick to the Euclidean distance, although we recommend experimenting with different loss functions depending on the system one is modelling. Any differentiable function $\mathcal{L} : \mathcal{X} \times \mathcal{X} \to \mathbb{R}$ can be used, if minimising its output constrains the inputs in a desirable way.

# 3 Results

We now demonstrate the applicability of the NCA to two contrasting use cases. First, we consider training data comprising numerical solutions of coupled nonlinear reaction-diffusion equations, with parameters that produce Turing patterns. Equations in this class are widely used to model complex biological systems such as in: developmental biology [33, 34]; ecology [35]; and skin pattern morphogenesis [36]. We adopt a representative example that is capable of generating a wide variety of spatial patterns, in the context of chemical reactions [17]. We demonstrate in particular that the NCA can generalise beyond the set of initial conditions in its training data in the context of a time series for which an underlying dynamics is known to exist. We then turn to an artificial image morphing problem, inspired by [11], and show that the same NCA is capable of constructing local rules to effect the desired dynamics. In contrast to the reaction-diffusion system, such rules are not known *a priori*, so it is not obvious that they exist. We also test the robustness of the rules that are learnt. When training to PDEs, the focus is on exploring training hyperparameters (time sampling) and sensitivity of training to

measurement noise in the data. After determining suitable training hyperparameters, we then show that this generalises to the image morphing task, where we explore model hyperparameters (kernels, activation functions, number of hidden channels) and the stability of trained models to perturbations of the input data.

### 3.1 Gray-Scott reaction diffusion equations

The Gray-Scott [17] reaction diffusion equations read

$$\partial_t A = D_A(\partial_{xx} + \partial_{yy})A - AB^2 + \alpha(1 - A)$$
$$\partial_t B = D_B(\partial_{xx} + \partial_{yy})B + AB^2 - (\gamma + \alpha)B$$

in which $D_A$, $D_B$, $\alpha$ and $\gamma$ are parameters. These describe two species, $A$ and $B$, which diffuse in two-dimensional space with diffusion constants $D_A$ and $D_B$, respectively. Species $A$ grows towards a density of $A = 1$ at a rate $\alpha$, whilst species $B$ dies out at rate $\gamma + \alpha$. The species also undergo the reaction $A + 2B \rightarrow 3B$.

With $D_A = 0.1$, $D_B = 0.05$, $\alpha = 0.06230$, $\gamma = 0.06268$ we obtain maze-like patterning (Fig 3). We solve these PDEs with an Euler finite-difference discretisation scheme, with time-step of 1, for $N = 1024$ steps, and on an $S \times S$ lattice of size $S = 64$ with periodic boundary conditions. $\alpha$ and $\gamma$ parameterise the patterning type, whereas $D_A$ and $D_B$ re-scale the patterns, and must be chosen in line with the timestep size to achieve numerical stability. Here we do not run the PDE trajectories long enough to reach a steady state, as we are more interested in the transient dynamics. For brevity we only show results with $\alpha = 0.06230$, $\gamma = 0.06268$, but we see similar performance on most other regions of parameter space where patterning occurs. The exception is that choices of $\alpha$ and $\gamma$ that give fast oscillations or unstable behaviours (rather than stable pattern formation) are harder to learn.

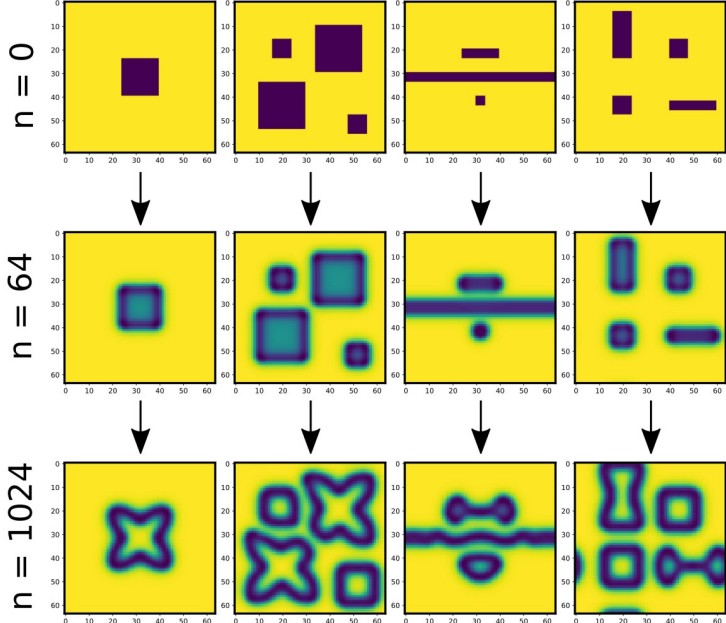

**Fig 3. Snapshots taken from the training data used for learning PDE dynamics.** PDE is run for $N = 1024$ steps with timestep 1 and $D_A = 0.1$, $D_B = 0.05$, $\alpha = 0.06230$, $\gamma = 0.06268$.

**3.1.1 Effect of training hyperparameters.** We begin with a basic NCA architecture that employs just $K = 2$ kernels, the identity and Laplacian, $p = 0$ (purely deterministic), $C = 8$ channels in total (so 2 observable and 6 hidden channels) and a rectified linear unit (relu, $u(z) = \frac{|z|+z}{2}$) activation function. We found that the Nadam optimiser [37] consistently performed well. Nadam is a modification of the widely used Adam optimiser [38], with the only difference being the use of Nesterov momentum [37]. Optimisers based on Nesterov momentum perform well, both in theoretical convergence and generalisability of trained deep neural networks [37, 39]. Note that we also employ gradient normalisation [40] before passing gradient information to the optimiser—this was found to significantly improve training performance. This just leaves the time sampling ($t$) as the main hyperparameter to optimise. Time sampling is subtle in the case of PDEs as numerical integration of the PDE system necessarily involves a discrete integration time step. We can sample the trajectories at coarser intervals, increasing $t$ in Algorithm 2 such that each NCA update corresponds to a timestep of the PDE solver. In other words, we only compare every $t^{\text{th}}$ PDE and NCA step.

We found that while training loss increased with greater sampling intervals $t$, tests on unseen initial conditions achieve comparable loss for most sampling intervals, with modest improvements for greater $t$ (Fig 4). The training loss is calculated for $N = 1024$ steps (as in Fig 3), whereas the test loss is calculated over $N = 2048$ steps from an unseen initial condition. This demonstrates generalisation both to unseen initial conditions, and longer simulation times. Note that the unseen initial condition used for testing features high frequency components not observed at all during training, and that NCA trained with high sampling were

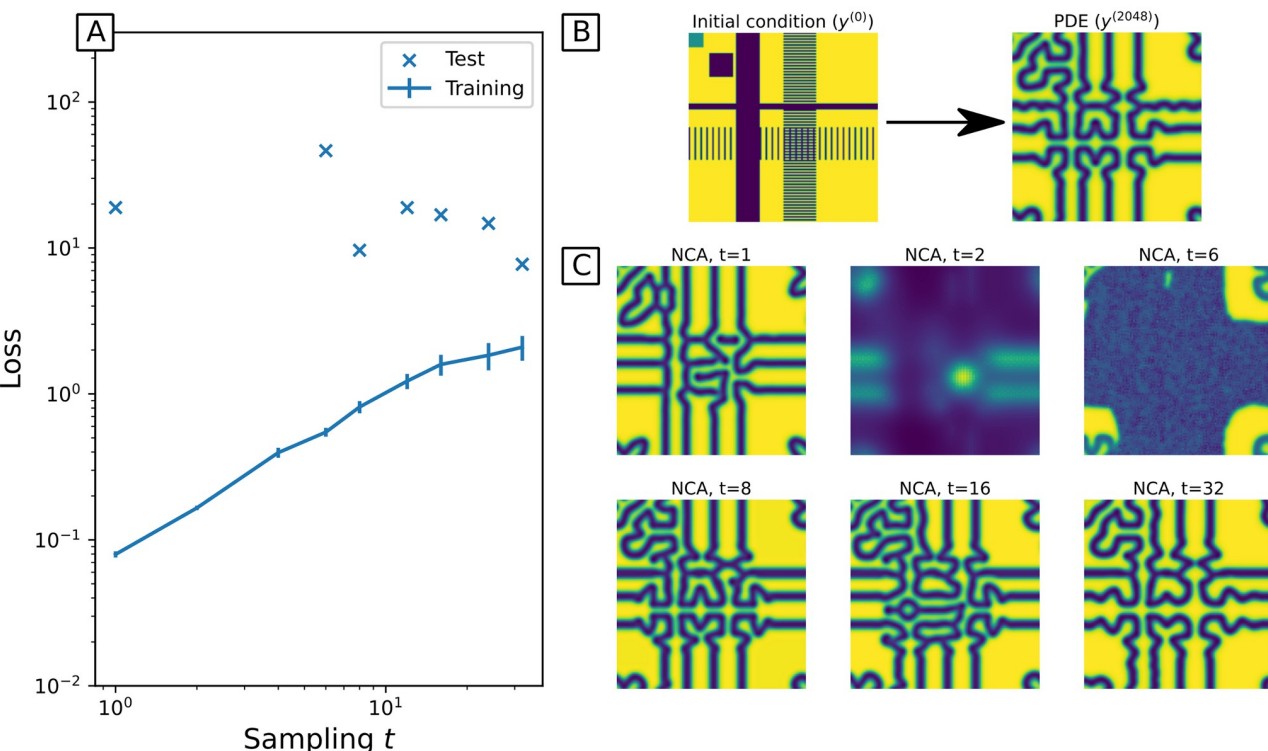

**Fig 4.** A: loss as a function of time sampling $t$. Training loss shows the minimum loss during training epochs, averaged over 4 random initialisations, with standard deviation as error bars. Test loss shows how the best trained NCA (minimal training loss) performs on unseen initial conditions. B: Initial condition and true state (PDE simulation) at $n = 2048$. C: Snapshots of NCA trajectories (at $n = 2048$) based on unseen initial conditions, with varying time sampling $t$. Each NCA is trained for 4000 epochs, with a mini-batch size $B = 64$.

sometimes numerically unstable to these high frequency inputs (missing test loss points in Fig 4A, or snapshots at $t = 2, t = 6$ in Fig 4C).

Fig 5 shows various snapshots from true (PDE) trajectories alongside the corresponding snapshots from an NCA trained with $t = 32$. This extrapolates an unseen initial condition far beyond the time observed during training ($n \in [0, 1024]$), demonstrating that the NCA does learn the underlying rules of the dynamics rather than overfitting to the training trajectories. When we considered finer sampling $t$ (Fig 4), we observe more frequent numerical instabilities, or complete failure to learn dynamics. Coarse time sampling appears to both stabilise these numerical problems, and yield more generalisable models. We posit this is due to coarse time sampling allowing the NCA to be less constrained during training, in that intermediate states may explore more possible states, increasing the chances of finding $\theta$ that gives the correct dynamics. Fine time sampling perhaps over-constrains the NCA, leading to instabilities or training converging to sub-optimal local minima of the loss landscape. Alternatively, the behaviour at fine time sampling is consistent with overfitting, so coarser time sampling could be considered a regularising technique here.

**3.1.2 Training on noisy data.** Unlike simulated data, experimentally-obtained training data is likely to have measurement noise associated with it. Whilst there are many techniques for cleaning and denoising data, we should still investigate how noisy data affect the training of NCA to reproduce patterning in the Gray-Scott model. We use training and model parameters that worked well on non-noisy data: an NCA with $C = 8$, relu activation and Identity and Laplacian kernels; trained with time sampling $t = 32$ for 4000 epochs with a Euclidean loss. Instead of training to PDE data, we now train to a noisy version as follows:

$$\tilde{y}_i = (1 - \xi)y_i + \xi\eta_i \qquad \eta_i \sim U(\min y, \max y)$$

Where $\xi$ interpolates between clean signal ($\xi = 0$) and pure noise ($\xi = 1$), and $\eta_i$ is drawn from a uniform distribution scaled by the maximum and minimum value of $y$ across the whole

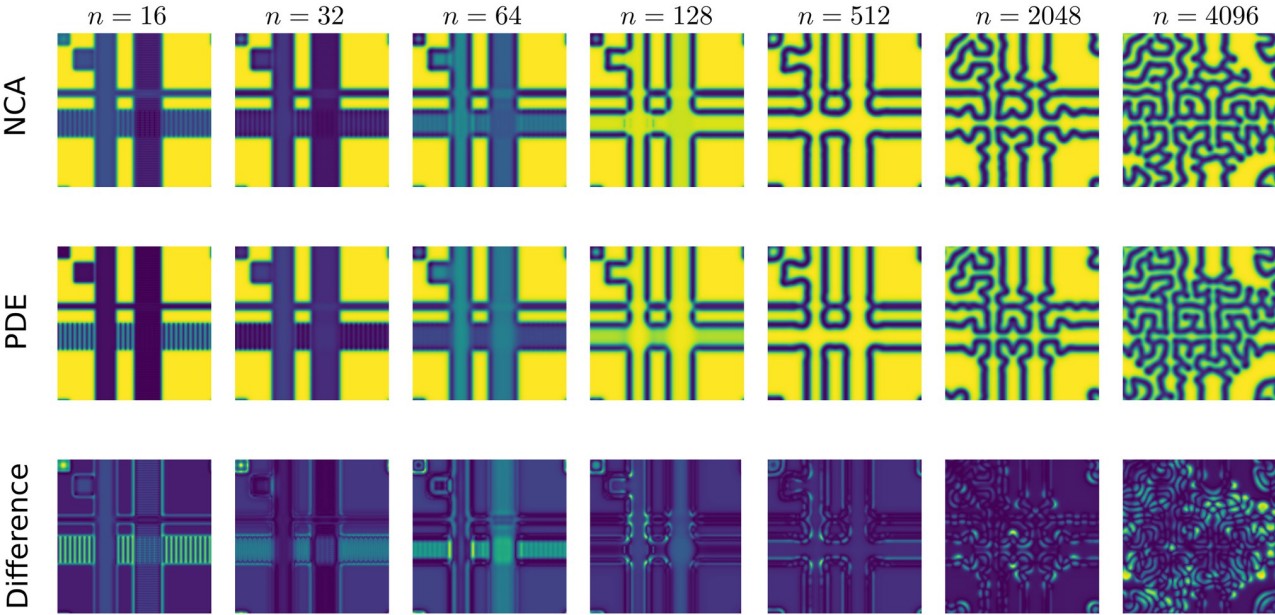

**Fig 5. Snapshots of PDE and NCA trajectories from an unseen initial condition.** NCA trained with $C = 8$, Identity and Laplacian kernels, relu activation, trained on sampling $t = 32$ for 4000 epochs with Euclidean loss.

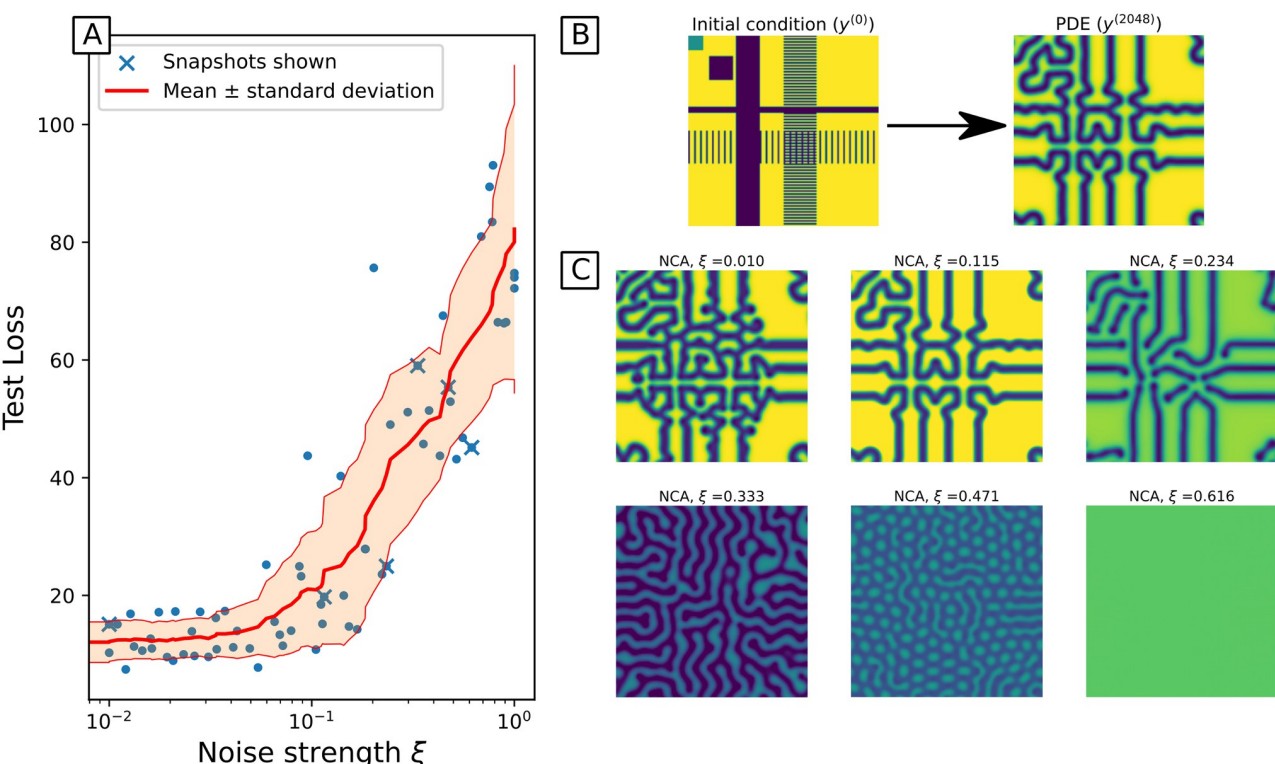

**Fig 6.** A: loss as a function of noise intensity $\xi$, on unseen initial condition. $\xi$ interpolates between the PDE trajectory ($\xi = 0$) and uniform noise ($\xi = 1$) as the training data for the NCA. Also shown is an interpolating moving average $\pm$ standard deviation, as there is significant variation introduced by random parameter initialisation. B: Initial condition and true state (PDE simulation) at $n = 2048$. C: Snapshots of NCA trajectories (at $n = 2048$) based on unseen initial conditions, with varying noise intensity $\xi$. Each NCA is trained for 4000 epochs, with a mini-batch size $B = 64$.

trajectory. As expected, for very small $\xi$ the models behave as in the noiseless case, reproducing the correct patterns (Fig 6). For large $\xi > 0.5$, we find that the NCA just learns a flat solution that smooths out the uniform noise. The interesting behaviour occurs when $0.2 < \xi < 0.5$, where the NCA learns some incorrect pattern formation consistent with other Turing patterns. This implies that with moderate noise, the NCA still learns some mechanisms for pattern formation, but not necessarily all of the correct ones. This can be interpreted as the NCA having a good inductive bias for learning Turing pattern forming systems, and this indicates the NCA are robust to noise up to $\xi = 0.2$.

In summary, we have found that the NCA architecture set out in Section 2 is capable of learning update rules that reproduce the solution of a certain pair of coupled nonlinear PDEs. Our main finding is that good rules can be learnt, but coarse time sampling improves numerical stability, and learns rules that generalise better to unseen initial conditions. We also find that the NCA is robust up to 20% measurement noise of the PDE system, but even when there is too much noise, it still learns mechanisms that form patterns, albeit incorrect ones.

## 3.2 Image morphing

We now test whether a training method that works well for PDEs also works on the image morphing task, where we do not know at the outset whether a local dynamics exists that is capable of transforming one image into the next, and if so, what form it takes. The task comprises an initial image, an intermediate image to morph through, and then a final image that is

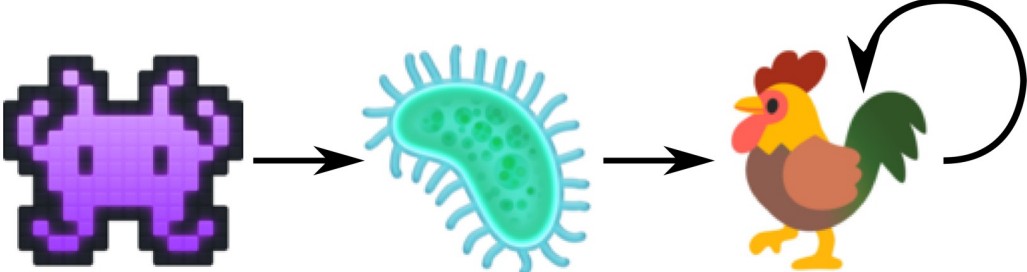

**Fig 7. Image morphing task.** Given a space invader initial condition, morph through a microbe and remain stable at a rooster pattern. Images taken from https://emojipedia.org/google, used under Apache License 2.0: https://github.com/googlefonts/noto-emoji/blob/main/LICENSE

intended to remain stable (i.e., be an attractor of the dynamics) (Fig 7). This latter requirement is incorporated into the training data by repeating the final image twice. The images shown were downsampled to a resolution of $60 \times 60$ to reduce computational cost. We further impose fixed boundary conditions, that is, to insist that the state vector $x^{(n)}$ vanishes at the boundary points. This is because the system is not periodic, as was the case for the PDE problem.

Reliable training of the NCA requires a careful construction of the training data. A side-effect of the fixed boundary conditions is that the NCA can learn to grow an image from the boundary, rather than from the initial condition. We would like the pattern formation to remain translationally invariant—the image morphing should behave independently of the boundaries, and should only depend on the input images. To enforce this, we embed the training data within a larger lattice, thereby reducing the influence of the boundaries. Translational invariance is further encouraged by training to several copies of the image sequence, each randomly shifted in space, to prevent learning any effective long range boundary interaction. It is possible to train NCA to produce textures as *global* attractors [13, 30] as textures are translationally invariant. We believe it is impossible to have a fixed pattern (i.e. not a translationaly invariant texture) as a global attractor—if boundary effects are removed so the whole NCA system is translationaly invariant. Avoiding the desired dynamics being a global attractor ensures that the NCA has learned a rule that maps the input state through the sequence of target states, rather than just generating the target states irrespective of initial conditions. This is further verified by exploring stability under perturbation in Section 3.2.2.

We also find that to train the NCA to reach a stable state, in effect mapping an image to itself, augmenting the training data with noise is necessary. Without noise augmentation, the training process crashes as gradients diverge (when training the final stable transition), but adding a very small amount of random noise to the data fixes this. We can understand the effect of this noise as introducing a basin of attraction around the desired final state, and training to noisy images enhances the robustness of the NCA to noisy perturbations.

**3.2.1 Effect of model hyperparameters.** The training hyperparameter $t$ that corresponds to the frequency of time sampling cannot be assumed to translate from the PDE case to the image morphing task. As there is no underlying physical mechanism connecting the images, there is nothing to provide a basic unit of time. We are however guided by the fact that the update rules are local, and therefore initially take the number of timesteps to be 64, which is similar to the lattice size and therefore gives sufficient time for information to propagate across it. As we explore this point more below, we find in fact that fewer timesteps can also be sufficient.

While a deterministic update ($p = 0$) is appropriate in the PDE case, as this was a feature of the training data, for the image morphing problem, we update stochastically with $p = \frac{1}{2}$, as removing global synchronisation between each cell acts like dropout regularisation. Note that a choice of $p = \frac{1}{2}$ effectively halves the number of timesteps between images. We found that varying the update probability $p$ had very little direct impact on model performance (except for extreme values close to 1 or 0), instead the effective number of timesteps between images was explored.

The system state has 4 observable channels—the red, green, blue and alpha (transparency) components—and 12 hidden channels. Since the images are not rotationally symmetric, and we have no prior underlying mechanisms to constrain symmetries of update rules, we consider adding gradient kernels to the identity and Laplacian kernels that were used in the PDE case. The Laplacian kernel detects spatial changes (and curvature) in patterns, whereas the gradient kernels also detect the orientation of any changes. We find that including the symmetry-breaking gradient kernels improves the performance over just using symmetric kernels (Fig 8A and

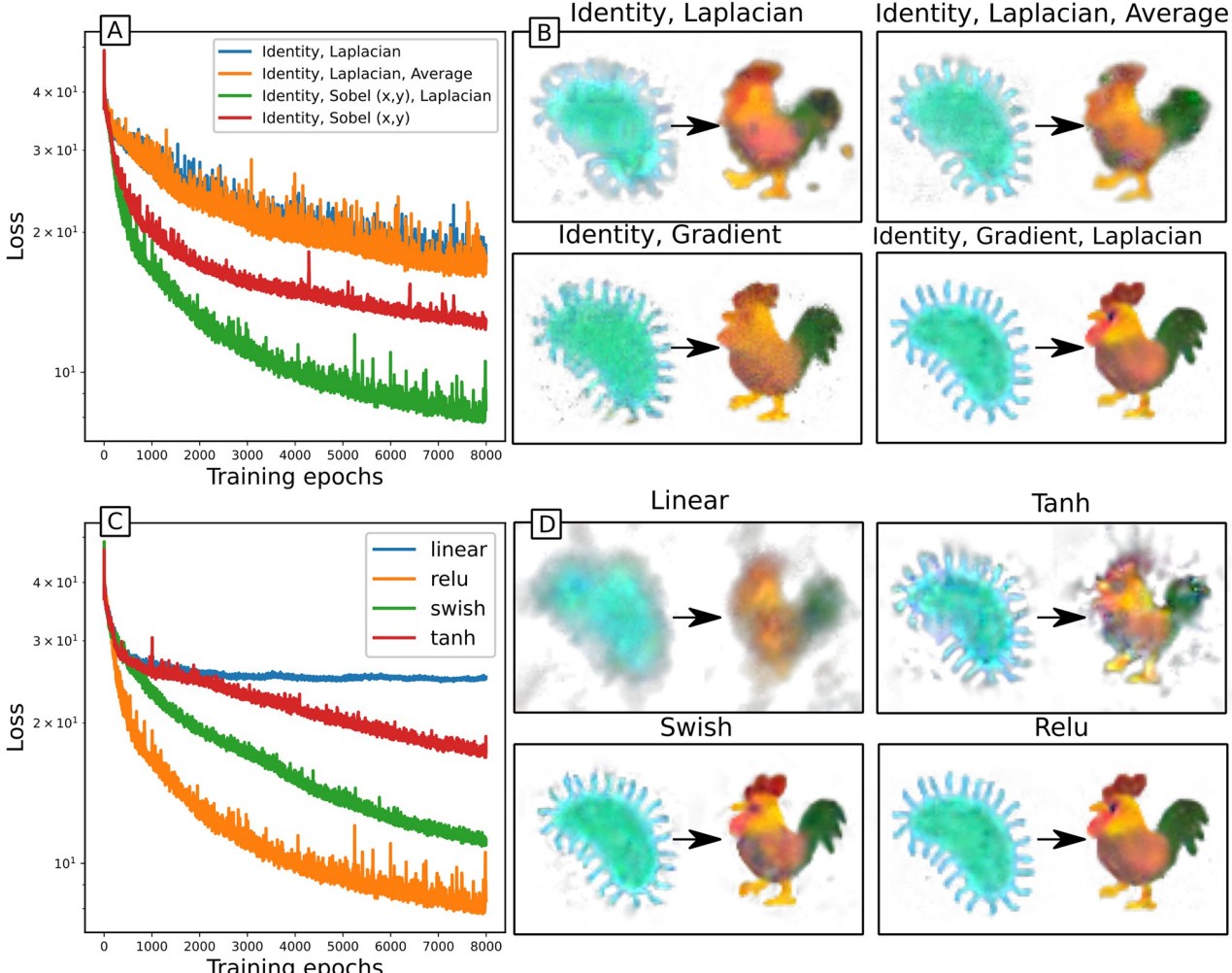

**Fig 8. NCA trained on image morphing task with different kernels and activations.** 16 channels, 64 steps between images. A,B: training loss and snapshots of NCA with relu activation and various kernels. C,D: training loss and snapshots of NCA with Identity, gradient and Laplacian kernels, for various activation functions.

8B). This does however break the symmetry of the NCA—such an NCA is unlikely to be stable under rotations or reflections of the initial condition, as discussed in Section 3.2.2. We also find that the best performing activation function is relu (Fig 8C and 8D). The linear case, $u(z) = z$, can be thought of as an absence of an activation function. Surprisingly, the overall shape of the final image is reasonably well reproduced, although it clearly lacks the definition achieved with the other activation functions. This justifies the additional complexity of nonlinear activations.

Exploring how NCA behaviour scales with number of channels, we find that unsurprisingly more hidden channels performs better (Fig 9A and 9B), capturing more of the details in the rooster image. The number of channels functions as a clear 'model size' parameter, and we find that model performance scales nicely with this measure of model size. We also explore how NCA train for different numbers of timesteps between images (Fig 9C and 9D). It is surprising that with as few as 8 timesteps, the basic shape and colour of the rooster are correct

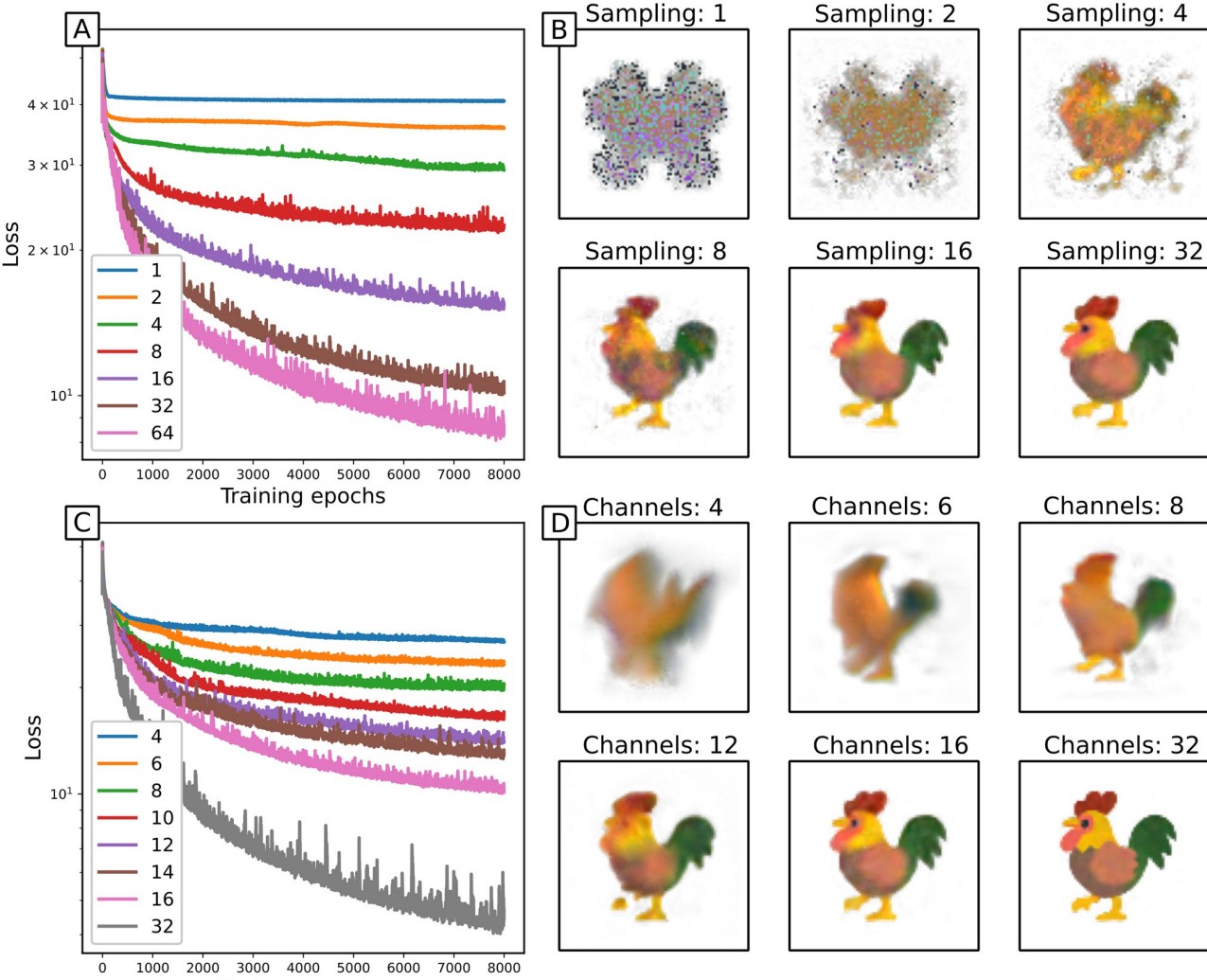

**Fig 9. NCA trained on image morphing task.** Relu activation; Identity, gradient and Laplacian kernels. A,B: Training loss and snapshots of 16 channel NCAs trained with different time sampling. C,D: Training loss and snapshots of NCAs trained with time sampling of 32, and various numbers of channels.

(although details are better at 16 or 32 steps), which highlights the locality of the update rule. The image resolution is $60 \times 60$, and with 8 timesteps between images only cells less than 16 pixels away can communicate (from initial condition to reaching the stable rooster pattern). However with the stochastic cell updates, the effective communication range from initial to final condition here is halved to just 8 pixels. This emphasises that the update rule is local, in that local structures of the initial condition morph into local structures of the final state at similar locations.

**3.2.2 Stability analysis.**   With a trained NCA, there is an obvious question of stability—if an initial condition is perturbed away from what the NCA is trained on, how does this affect the behaviour? We consider three kinds of perturbations of the initial condition: local perturbations, global perturbations, and symmetry perturbations. Local perturbations change one pixel, and allow us to explore how errors propagate through the spatial part of the NCA. Global perturbations can show how resilient NCA are to noisy inputs. Symmetry perturbations, such as rotations or reflections of initial conditions, allow us to explore how NCA respect desirable symmetries.

Stability under local perturbations depends strongly on how many timesteps between images the NCA is trained on (Fig 10). We find that NCAs with fewer time-steps are more stable to local perturbations, or conversely that allowing more NCA steps between training images gives more time for local perturbations to travel. In both cases the perturbations remain mostly local. Using local perturbations could help calibrate the number of timesteps to use when modelling real systems, in that the NCA should have the same response to perturbations as the underlying data being modelled.

To address the question of stability with respect to global perturbations, we frame it as an optimisation problem. Let $\kappa^n(x^{(0)}, \tilde{x}^{(0)}) = \| \tilde{x}^{(0)} \| - \| \Phi^n(x^{(0)} + \tilde{x}^{(0)}) - \Phi^n(x^{(0)}) \|$, where $\tilde{x}^{(0)}$ is a perturbation of initial condition $x^{(0)}$. By finding $\tilde{x}^{(0)}$ that maximises or minimises $\kappa^n$, we can find a maximally perturbed initial condition $x^{(0)} + \tilde{x}^{(0)}$ that leaves a future state $x^{(n)}$ unchanged, or a minimal perturbation that destroys $x^{(n)}$. This allows us to explore the space of

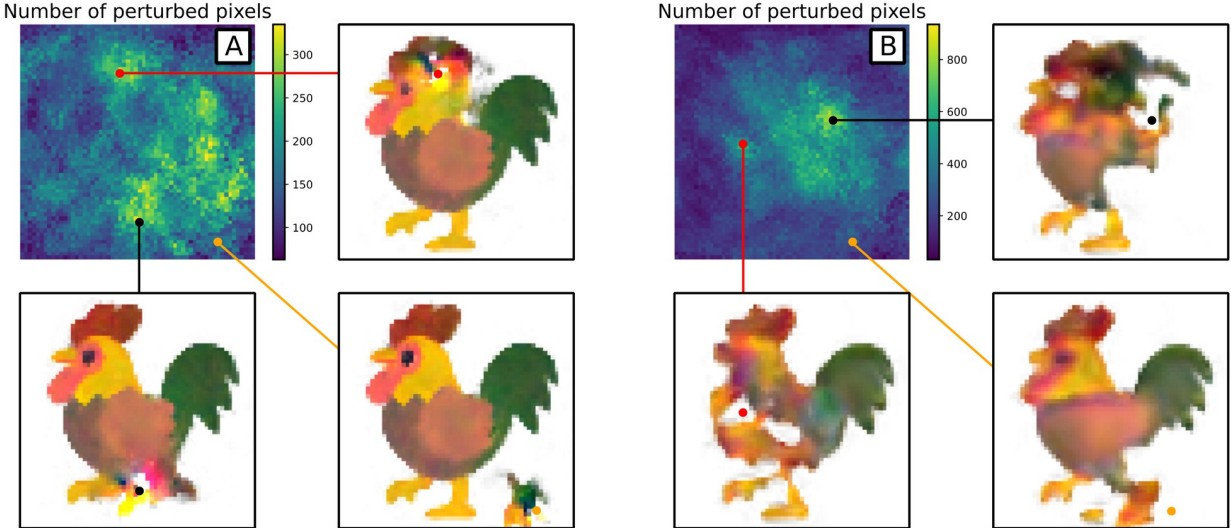

**Fig 10. Local stability behaviour of two NCA.** A: 32 channels, 32 steps between images. B: 16 channels, 64 steps between images. Top left heatmap in each case shows how many pixels of the final image change (by more than 0.1 to account for random fluctuations) when that pixel is perturbed in the initial condition. The other images all show snapshots of the final state when the initial condition is perturbed locally, for different perturbation locations.

initial conditions around which the NCA was trained, and can reveal which features of an initial condition are important. For example, it may be only the edges and corners of an image are learned from. As the whole NCA process is differentiable, we can use gradient based optimisation on $\kappa^n$ to find $\tilde{x}^{(0)}$. Finding minimal perturbations that destroy $x^{(n)}$ is similar to adversarial attacks of classifier networks, where small changes to an image completely destroy the behaviour of an image classifier. Fig 11 shows the behaviour of a trained NCA (best performing model shown in Fig 9A and 9B) starting from examples of these adversarial initial conditions. It is possible to find initial conditions that are visually similar to the true initial condition, and yet they destroy the stable rooster pattern ($x^{(96)}$). We can also find large perturbations of the initial condition that leave the target state ($n = 96$) unperturbed, however the long term stability of the rooster pattern is still damaged.

Finally, we compare the behaviour of different NCA models on symmetrically perturbed initial conditions (Fig 12). By rotating or flipping the input image we obtain symmetrically perturbed inputs. One NCA is trained on *normal data*, that is, data that has only been translationally perturbed to minimise boundary effects. The other is trained on *augmented data* that also includes the same training data after applying global rotations about random angles. We also explore the effect of restricting the NCA to include only symmetric kernels, rather than both symmetric and asymmetric kernels. We find that even without any data augmentation, the symmetric kernel NCA already performs very well, although it struggles with the off lattice 45 degree rotations. When trained on rotationally augmented data, the asymmetric kernel NCA improves its performance on rotated inputs, but is still outperformed by the symmetric kernel NCA for on-lattice rotations (with or without data augmentation). Off lattice rotations

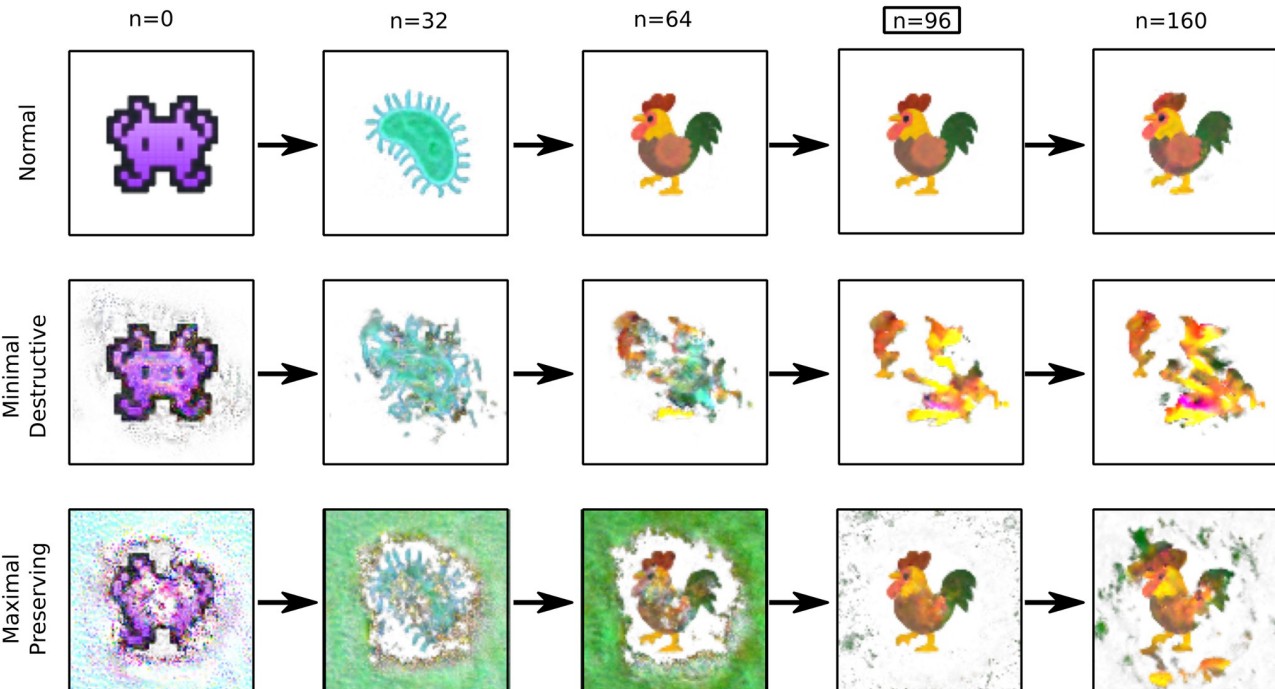

**Fig 11. Rightmost column shows extrapolation beyond training time, demonstrating stability of the final state.** Top row shows snapshots from unperturbed trajectory. Middle row shows snapshots from minimal initial perturbation that destroys the final state (minimising $\kappa^{(96)}(x^{(0)}, \tilde{x}^{(0)})$). Bottom row shows snapshots from maximal initial perturbation that preserves the final state (maximising $\kappa^{(96)}(x^{(0)}, \tilde{x}^{(0)})$). NCA (32 channels; Identity, gradient and Laplacian kernels; time sampling $t = 32$, relu activation) trained on image morphing task. Initial condition taken from https://emojipedia.org/google, used under Apache License 2.0.

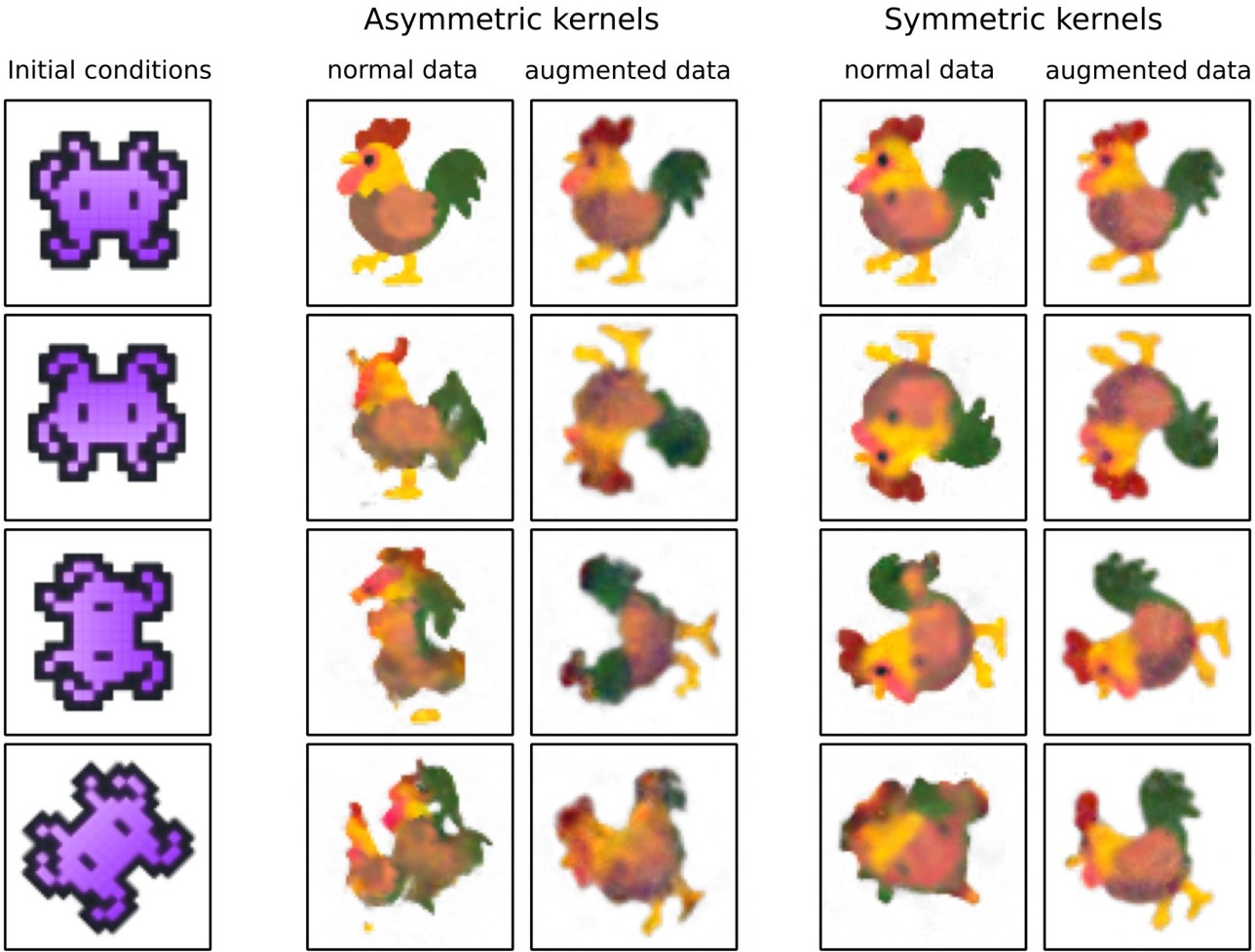

**Fig 12. Behaviour of trained NCA on symmetrically perturbed inputs.** Left column shows inputs, middle two shows final state behaviour for NCA with asymmetric kernels (identity, gradient and Laplacian), rightmost two shows final state behaviour for NCA with symmetric kernels (identity, Laplacian, average). Augmented data examples show NCAs trained to trajectories rotated to random angles. Initial condition taken from https://emojipedia.org/google, used under Apache License 2.0.

are the most challenging, and seem to be where data augmentation is necessary even for symmetric kernel NCA. Overall, we find that the symmetric kernel NCA better handles symmetric perturbations, whilst the asymmetric kernel NCA performs best on the unperturbed data. As one might expect, building symmetry into the model allows it to solve a broader range of symmetrically-related problems, whereas leaving it out promotes specialisation towards a single problem.

## 4 Discussion

We have demonstrated NCA as a framework for modelling emergent spatio-temporal patterning. Many systems in biology are characterised by complex emergent phenomena of locally interacting components, and finding interaction rules or mechanisms that lead to specific emergent behaviours is a challenging inverse problem. By making classic cellular automata differentiable, NCA present a new approach to this class of problems. [11] demonstrated that a

trained NCA can generate complex structures (specifically emojis) that remain stable over time and to perturbation. Here we have extended this approach to learn dynamics from snapshots of a pattern at multiple timepoints (rather than just the end-point), i.e., we show the ability to learn dynamic patterns, specifically those arising from PDEs with Turing instabilities that been widely used to study biological pattern formation.

Specifically, we showed in Section 3.1 that NCA can infer update rules equivalent to those obtained by discretising and iterating a set of PDEs. NCA have an inductive bias to learning underlying dynamics, rather than just overfitting to data, due to their minimal parameters and hard coded local kernels. We demonstrate this by presenting the trained NCA with initial conditions that were not part of the training data, and finding that the predicted trajectories are similar to those obtained directly from the PDEs—the trained NCA generalise well. This suggests that an NCA trained on experimental data could be used to predict the behaviour of that system under conditions that have not been directly observed. We have also discussed NCA hyperparameters in more detail than most previous work, which can provide guidance for future exploration. For example, tuning the number of timesteps between images can constrain how far local information is allowed to spread, and the number of channels required to accurately capture a patterning behaviour could function as a heuristic for the complexity of that pattern.

More generally, the findings of Section 3.2 confirm that NCA can be used as a tool to construct local dynamical update rules whose emergent properties satisfy certain constraints. These constraints include the end state, but extend also to the stability of that configuration, invariance of the dynamics under certain symmetry operations and the effect of boundary conditions. Given that for any observed emergent patterning, many possible microscopic update rules could exist, constraining how NCA respect symmetries or behave around boundaries helps reduce the set of possible rules. As the training process for NCA amounts to a differential optimisation procedure, microscopic rules that yield desired emergent behaviour can be efficiently found, even when constraints are imposed, as exemplified by the image morphing task. It is worth emphasising the non-uniqueness of these microscopic rules, which occur on multiple levels. Firstly, if there is no assignment of meaning to hidden channels, different NCA can learn to use different permutations of hidden channels arbitrarily. Secondly, the internal hidden layer and linear output layer have a similar permutation symmetry. More generally, beyond simple permutations, we find that different initialisations lead to different nonlinear combinations (hidden weights) beyond a simple permutation that lead to the same macroscopic dynamics. This is in line with similar results for Turing pattern forming PDEs [41]. Non-uniqueness of rules is not itself a problem, but will need to be considered when analysing or interpreting trained NCA.

Using NCA as an *in silico* way to study the behaviour of growing systems is a recurring theme in the literature [11, 21], discussed more concretely in [42]. Of particular biological interest is the potential for modelling regrowth or tissue repair, alluded to in [11, 21], which has to our knowledge not yet been applied to real biological data. NCA have been trained to exhibit regenerative properties on the toy problems of emoji growth from a single pixel [11], and texture formation from noise [30]. In both cases, if large patches of the NCA state are reset to their initial state (zero for emoji growth, noise for texture formation), the stable pattern regrows from the remaining sections. The technique for achieving regeneration here is largely in the training process, where NCA are explicitly trained to repair the target image when 'damaged' during training. These are interesting models of regenerative pattern formation, but have not yet been constrained to measurements of biological systems. Further applications or extensions of NCA models have been explored in the context of image processing and synthesis [43, 44]. Here NCA models have been coupled to CLIP text embeddings [45], in line with

recent machine learning based text to image techniques. NCA are clearly capable of a diverse range of behaviours, but all the previous literature just focuses on training one set of transitions, rather than full dynamic trajectories. We believe that the extension to learning dynamics of arbitrarily long sequences dramatically increases the already wide range of systems and behaviours NCA can model. Specifically, being able to train to sequences of images should better enable NCA to model real biological systems, such as regeneration and tissue repair, as well as enabling more interesting image synthesis techniques.

Compared to most current machine-learning research, our chosen neural network architecture is economical in terms of the number of trainable parameters. This not only makes training more computationally efficient, but also adheres also to the aesthetic guidance of modelling traditions in physics and mathematics that simpler models are preferred over more complex models when they have comparable descriptive power. Here we have found that a single hidden layer is sufficient to model a variety of systems and reproduce a wide range of behaviours. The reason for this may lie in part due to hidden channels in the state space, as these can encode complex environmental information such as boundary conditions, as well as encoding memory for longer term dynamics. This spatially distributed memory encoding in the hidden channels could be likened to previous work on differentiable neural computers [46]. We note that NCA also link back to older work on amorphous computing [47], providing a connection between modern machine learning and theory of spatially distributed computational frameworks.

There are however a few shortcomings of NCA, mainly the underlying assumption that purely local interactions are sufficient. There will be systems with non-local (or multiscale) interactions that cannot be elegantly explained with purely local rules. We have also assumed that the update rules are constant over time, even though many complex (physical or biological) systems are highly time dependent [48]. Whilst it is possible that the hidden channels in the NCA could encode nonlocality and time-dependence, it might be more natural (and interpretable) to extend the representation of the dynamics in the neural network to incorporate such dependencies explicitly, for example by increasing the size/stride of convolution kernels, or by including an explicit time parameter as a network input. A possible risk with including explicit time dependence is that the NCA could over-fit to the timestep, rather than learning the microscopic interactions that yield the emergent behaviour. To tackle such questions, it might be desirable to augment the loss functions with measures of model complexity as a means to converge on the most parsimonious description, for example by sparsity regularisation. Generalising NCA to better work on multiscale systems could also be worth exploring, for example by coupling NCA-like models on lattices of different resolutions.

Similarities can be drawn between NCA and other data driven equation discovery techniques like SINDy [49] or Extended Dynamic Mode Decomposition (EDMD) [50]. Both SINDy and EDMD have the main purpose of fitting dynamical systems to data, both in the cases of ODEs and PDEs. SINDy enforces parsimonious solutions through sparsity regularisation, whereas Dynamic Mode Decomposition is analogous to Singular Value Decomposition for time series data (and the Extended DMD is a nonlinear generalisation). The key areas where NCA differ is in background motivation, and the sorts of systems they're suited to. NCA act as a bridge between machine learning based image processing, and learning simple models of complex systems; whereas SINDy and EDMD were developed in the context of data driven engineering and fluid dynamics respectively. NCA have strong performance on pattern formation, and are also suited to efficiently learning long term dependencies from sparsely sampled data. SINDy is designed for ODEs and PDEs in general, however it requires measurements of derivatives of the training data, and high time resolution. EDMD also requires high time resolution data, and is typically used in fluid dynamics. Cellular automata models are

more general than (local) PDEs. Although we restrict ourselves to differential operator kernels, we don't have to—learning arbitrary kernels would provide far more general expressive power, whilst still keeping a minimal (and importantly local) model. The class of models that could be learned with arbitrary kernels includes local PDEs, but is far more general.

A further development of these NCA models is to make them truly distributed during learning. When computing the loss, information of the whole lattice is needed, which places limits on the size of a lattice that can be handled with available computation time and memory. If instead an NCA could be trained with a purely local loss, such that model weights are updated for each pixel based on its neighbours, more advanced training procedures could be exploited. In essence, if the only global communication is the updates to the model weights, rather than the full lattice state, NCA could be trained using online training, or on much larger lattice sizes. An alternative approach to increase the resolution that can be trained on would be to randomly sub-sample elements [51] of the NCA lattice when computing losses.

The minimalism of the NCA model is chosen to facilitate interpretability. We believe this gives NCA major potential over larger neural network structure for learning dynamics that are explainable. Although fully analysing the trained models is still a challenge for future work, even with just one hidden layer, it is a far more tractable problem than analysing the larger neural networks that are commonly used [23]. It is still worth outlining a few potential directions: by directly reading the model weights on the input layer, we can assign importance of each kernel to each channel—this allows us to gain insight into what each channel may actually represent. For example, if one channel is predominantly read by the Laplacian kernel, it may be driving diffusive behaviour. We can also try to measure the 'importance' of each channel—if we train using weight decay, channels associated with large input weights are likely crucial to the dynamics, whereas those with weights close to zero are less so. If we have a lot of channels with almost zero weights, this may indicate that fewer channels are needed to represent the dynamics. Although we explored the stability of NCA under perturbations to the trajectory (initial condition), we did not address stability under perturbation of model parameters. This could naturally tie in to interpretation of the trained model, for example, by assessing stability under perturbation or dropout of network weights. It is likely that these models are not particularly stable under perturbations to their parameters—an interesting approach would be to enforce stable model parameters during training, with something like sharpness-aware minimization [52]. The recently popular field of explainable AI [14, 15] likely offers some other tools that would enable understanding trained NCA. It is also worth stating that interpretability of NCA could be addressed during the training process—for example if we are training to a set of trajectories that are parameterised in some way, we can encode dependence on this parameter into the neural network. This may be especially suitable to biological applications, where we could be modelling how an observed pattern formation depends on the concentration of some introduced or inhibited signalling molecules.

There is also potential for constructing even simpler models with comparable performance. Training with network pruning [53] could be a powerful approach, yielding more sparse models. This would give smaller and more interpretable models. Alternatively we could use spectral techniques [54]—by enforcing sparsity in the singular value decomposition (SVD) representation of model parameters. The SVD representation may also give some insight to model stability to perturbation—specifically which channels are likely to grow exponentially if perturbed from what is seen during training. A worthwhile further development would be to reverse-engineer a concise analytic expression for the underlying PDE from the trained NCA parameters, for example with symbolic regression techniques like Sparse Identification of Nonlinear Dynamics (SINDy) [49]. Such an approach could be tested with NCAs trained on known

PDEs, but it would be interesting to then apply this to systems where we don't know the underlying mechanics, such as image morphing or biological data.

## Supporting information

**S1 Appendix. See Supplementary Information section 1 for full gradient calculation.**
(PDF)

**S1 Video. A trajectory of an NCA with: 16 channels; identity, Sobel and Laplacian kernels; relu activation, trained on the image morphing task. Annotated with timestep $n$ in the top left corner.** Initial condition taken from https://emojipedia.org/google, used under Apache License 2.0.
(MP4)

**S2 Video. A trajectory of the same NCA, but with hidden channels displayed as RGB in the top right, bottom left and bottom right quadrants.** Observable channels displayed in the top left quadrant. Annotated with timestep $n$ in the top left corner. Initial condition taken from https://emojipedia.org/google, used under Apache License 2.0.
(MP4)

**S3 Video. Observable and hidden channels of an NCA trajectory, but this NCA has no activation function (i.e. is linear). Annotated with timestep $n$ in the top left corner.** Initial condition taken from https://emojipedia.org/google, used under Apache License 2.0.
(MP4)

**S4 Video. PDE trajectory corresponding to Fig 5.** Annotated with timestep $n$ in the top left corner.
(MP4)

**S5 Video. NCA trajectory corresponding to Fig 5.** Annotated with timestep $n$ in the top left corner.
(MP4)

**S6 Video. Difference between NCA and PDE trajectories in Fig 5.** Annotated with timestep $n$ in the top left corner.
(MP4)

## Acknowledgments

This work has made use of the resources provided by the Edinburgh Compute and Data Facility (ECDF) (http://www.ecdf.ed.ac.uk/). For the purpose of open access, the author has applied a Creative Commons Attribution (CC BY) licence to any Author Accepted Manuscript version arising from this submission.

## Author Contributions

**Conceptualization:** Alex D. Richardson, Tibor Antal, Richard A. Blythe, Linus J. Schumacher.

**Formal analysis:** Alex D. Richardson.

**Investigation:** Alex D. Richardson.

**Methodology:** Alex D. Richardson.

**Project administration:** Tibor Antal, Richard A. Blythe, Linus J. Schumacher.

**Resources:** Linus J. Schumacher.

**Software:** Alex D. Richardson.

**Supervision:** Tibor Antal, Richard A. Blythe, Linus J. Schumacher.

**Visualization:** Alex D. Richardson.

**Writing – original draft:** Alex D. Richardson, Tibor Antal, Richard A. Blythe, Linus J. Schumacher.

**Writing – review & editing:** Alex D. Richardson, Tibor Antal, Richard A. Blythe, Linus J. Schumacher.

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
