## [Decision Letter · Decision Letter 0]

6 Feb 2024

Dear Mr Richardson,

Thank you very much for submitting your manuscript "Learning spatio-temporal patterns with Neural Cellular Automata" for consideration at PLOS Computational Biology.

As with all papers reviewed by the journal, your manuscript was reviewed by members of the editorial board and by several independent reviewers. In light of the reviews (below this email), we would like to invite the resubmission of a significantly-revised version that takes into account the reviewers' comments.

We cannot make any decision about publication until we have seen the revised manuscript and your response to the reviewers' comments. Your revised manuscript is also likely to be sent to reviewers for further evaluation.

Sincerely,

Julio R. Banga, Ph.D.

Guest Editor

PLOS Computational Biology

Pedro Mendes

Section Editor

PLOS Computational Biology

I have completed my evaluation of your manuscript. The reviewers recommend reconsideration of your manuscript following major revision. Please resubmit your manuscript after addressing the comments below.

Reviewer's Responses to Questions

**Comments to the Authors:**

Reviewer #1: The paper presents an interesting approach to modelling emergent spatio-temporal patterns using Neural Cellular Automata (NCA). The authors show that NCA can learn dynamics from snapshots of patterns at multiple timepoints and can infer update rules equivalent to those obtained by discretizing and iterating partial differential equations (PDEs) commonly used in biological pattern formation.

This is an excellent paper and I am excited about the future work that it can lead to.

The only errors I can find are typographical. At the end of page 12, where "it's" should be "its" and in the caption of fig 8 C and D are separated by a period, rather than a comma.

I am also very grateful for the authors releasing the code and I hope people take it up.

The rest of my comments are either questions, or requests, for further clarifications.

1) During the simulations (fig 3, for example) generating learning data the final time step does not appear to be a steady state. Does this influence the learning of the NCA?

2) Figure 4b confuses me. The times are mixed with PDE results and are in a grid rather than given as a line, as in fig 5, which is much clearer.

3) Could the authors add a paragraph to the discussion regarding the potential of their work? Namely, I assume the overall future aim of this work would be to make the rules that the NCA learn to be interpretable; can we understand what the NCAs are doing, rather than using a blackbox neural network? Moreover, how unique are the rules that the NCA learn? The links in with the recent paper

Woolley, T.E., Krause, A.L. & Gaffney, E.A. Bespoke Turing Systems. Bull Math Biol 83, 41 (2021). https://doi.org/10.1007/s11538-021-00870-y

which shows that Turing systems are generally underconstrained so there are infinite families of systems that will produce similar patterns.

Reviewer #2: In this paper, the authors present an advancement in the field of Neural Cellular Automata (NCA), with the aim of bridging machine learning and mechanistic modelling. To do so, the authors have trained NCA to decipher complex dynamics from image time series and PDE trajectories, focusing on uncovering local rules that dictate large-scale dynamic and emergent behaviors. Notably, they expand the application of NCA beyond the conventional realm of learning stationary emergent structures. The paper demonstrates the generalization capabilities of NCA beyond PDE training data, and it offers insightful exploration into constraining NCA to adhere to specific symmetries, and the impacts of hyperparameters. I particularly appreciated the clear explanation of the atrchitecture of the network and its training process, a detail often omitted in AI literature. This clarity makes the paper accessible to a wide audience. However, the paper's primary limitation is its limited alignment with computational biology, especially in the image morphing section. Additional comments are provided below.

- Major comments

1) As explained in the introduction, one of the motivations of learning the rules of cellular automatas is solving the inverse problem of learning the rules behind a biophysical process. This is pursued further in the network architecture trying to generate a network that can be interpreted easily. Nevertheless, the authors did not analyze the resulting trained network e.g. what can recover/learn about the PDE once the network is trained? I can't think of any biological question in which I would care about the future dynamics of an image without being interested in extracting the mechanistic process underneath. Questions that would be interesting are:

1a) Can the diffusion rates be inferred from the training?

1b) Does the structure of the hidden layers give clues on the first and third order reactions?

1c) Are their parameter sets for which the training fails or is less informative?

2) As explained in the second part of the manuscript, it is interesting to understand predictions under noise perturbations. This is also true for the PDE problem since experimental data will always contain some noise. How robust is the PDE training to noisy data? My guess is that it should work, but it would be good to see it in action as in the image morphing example.

3) There are some comments in the discussion about the comparison of NCA with other similar AI approaches for PDE inference such as SINDy, but I found they were a bit limited. Since this paper shows a method that connects AI learning to PDE inference, I would like to know more about how these approaches compare. What are the limitations of both? What problems would be more suitable for one or the other? Are there any difference in the training constrains?

4) I did not understand the biological relevance of the image morphing study. I appreciate that the authors chose a simplistic emoji problem to make the analysis clear and general. Nevertheless, a significant text is missing supporting the motivation of this problem in a computational biology context.

- Minor comments

1) Along the paper, Sobel kernels are used as a synonym of gradient. But Sobel kernels also generate some smoothing of the data. This is not clear when they are connected to PDE operators or biophysical processes. I don't think this is a problem in the training, as the gradient operator can probably be expressed as a combination of the 5 kernels used. This should be clarified.

2) The description of the training was very clear, and I really appreciate the effort the authors put into writing this section. There is a minor abuse of notation in the inisialisation of the variables and comparing the loss - "We initialise x(delta t) = y(delta)" vs "we compare x(delta t) to y(delta)"). Perhaps it would be good to indicate the initial points with a hat as it is done in Algorithm 2?

3) I also found a bit confusing the \\delta notation e.g. y(\\delta) vs x(\\delta t). I can't think of a better candidate right now, but for me it broke a bit the reading flow of the text, for instance when interpreting Figure 2.

4) Figs. 4 and 5 could be clearer. On one hand 4B could reorganize the panels to make clear the order/hierarchy of the images. It would also be good to have some spatial difference (or loss) between the PDE and the different NCA solutions, so it is more evident which parts of the solution are not well recovered. Similar thing for Figure 5.

5) In 3.1 "DA and DB re-scale the patterns, and must be chosen in line with the timestep size to achieve numerical stability". Should not the timestep be chosen according to diffusion rate, not the other way around?

6) Boundary conditions for the PDE are periodic, but this is not mentioned until the image morphing section.

7) This comment is related to one of my major comments. In section 3.2.1 "removing global synchronisation between each cell acts like dropout regularisation, and can be biologically motivated". I failed to understand this comment, since there was no reference to biology prior to this comment in the image morphing scenario.

Reviewer #3: Referee Report for "Learning spatio-temporal patterns with neural cellular automata"

Overview: The manuscript by Richardson et al. explores the extension of cellular automata approaches, integrating neural network architectures to learn local rules from dynamic and transient states, as opposed to static images. This novel approach is demonstrated through two applications: learning Turing patterns from PDE outputs and transitioning between images using emojis. The methodology, grounded in a minimalist design with a single hidden layer, is both innovative and intriguing. The paper also examines the stability of these models against variations in initial conditions. While the work is technically robust and presents an interesting concept, there are aspects that could be improved to align more closely with the scope and expectations of PLoS Computational Biology.

Comments and Suggestions:

1. Technical Focus and Readability:

The manuscript is heavily technical, with even the Results section focusing primarily on methods. While the methods are thoroughly and clearly explained, this technical orientation may be more suitable for a specialized journal. Consider revising to include more application-focused results or discussions to broaden the appeal for a PLoS CB audience.

Suggestion: Enhance Fig. 1 by incorporating numbered steps in the schematic and a corresponding detailed explanation. This would aid in making the figure more accessible and self-explanatory to readers who may not be deeply familiar with the technical aspects.

2. Biological Relevance:

The connection to biological systems and processes is not strongly articulated (emojis are not sufficient). PLoS CB emphasizes biological insights, and this paper could benefit from a clearer exposition on this front.

Suggestion: Expand the discussion on how the local rules derived from the model correlate with biological phenomena, such as self-repair or regeneration. Are there implications or insights that this model provides in understanding these biological processes?

3. Explanation of PDE Correspondence:

The paper mentions that the rules derived correspond to discretized solutions of PDEs, but this aspect is not elaborated sufficiently.

Suggestion: Provide a more detailed explanation or a dedicated section on how these rules relate to the discretized solutions of PDEs, and whether this draws parallels with Turing's original models of discrete cellular systems using hopping instead of diffusion.

4. Parameter Sensitivity Analysis:

An investigation into the sensitivity of the model to its parameters is absent. Such an analysis could be crucial, especially in the context of robustness and biological relevance.

Suggestion: Conduct a sensitivity analysis to understand how variations in parameters affect the model's output. This would not only enhance the robustness of the model but also provide insights into the robustness problem in Turing systems.

Conclusion:

The paper by Richardson presents a novel and technically sound approach to learning spatio-temporal patterns using neural cellular automata. However, to align more closely with the readership and scope of PLoS Computational Biology, it would benefit from significant enhancements in readability, biological relevance, detailed explanations of key concepts, and robustness analysis. These improvements could significantly elevate the impact and applicability of the research within the field.

**Have the authors made all data and (if applicable) computational code underlying the findings in their manuscript fully available?**

Reviewer #1: Yes

Reviewer #2: Yes

Reviewer #3: Yes

PLOS authors have the option to publish the peer review history of their article (what does this mean?). If published, this will include your full peer review and any attached files.

Reviewer #1: **Yes: **Thomas E. Woolley

Reviewer #2: No

Reviewer #3: No

Figure Files:

Data Requirements:

Please note that, as a condition of publication, PLOS' data policy requires that you make available all data used to draw the conclusions outlined in your manuscript. D

---

## [Decision Letter · Decision Letter 1]

2 Apr 2024

Dear Mr Richardson,

We are pleased to inform you that your manuscript 'Learning spatio-temporal patterns with Neural Cellular Automata' has been provisionally accepted for publication in PLOS Computational Biology.

Best regards,

Julio R. Banga, Ph.D.

Guest Editor

PLOS Computational Biology

Stacey Finley

Section Editor

PLOS Computational Biology

Reviewer's Responses to Questions

**Comments to the Authors:**

Reviewer #1: The authors have updated the manuscript and clarified the points that I have raised.

Reviewer #2: The authors have successfully addressed all my questions and the questions from the other referees.

Reviewer #3: The revised version is significantly improved and hence I suggest publication. Although it is still very technical and blackbox-like, and the biological implications are limited, the new section on training on noisy data, the discussion on biological relevance, and the the additional explanations throughout the text make this an important contribution.

**Have the authors made all data and (if applicable) computational code underlying the findings in their manuscript fully available?**

Reviewer #1: Yes

Reviewer #2: Yes

Reviewer #3: Yes

PLOS authors have the option to publish the peer review history of their article (what does this mean?). If published, this will include your full peer review and any attached files.

Reviewer #1: **Yes: **Thomas Woolley

Reviewer #2: **Yes: **Ruben Perez-Carrasco

Reviewer #3: No

---

## [Editor Report · Acceptance letter]

19 Apr 2024

PCOMPBIOL-D-23-01636R1 

Learning spatio-temporal patterns with Neural Cellular Automata

Dear Dr Richardson,

I am pleased to inform you that your manuscript has been formally accepted for publication in PLOS Computational Biology. Your manuscript is now with our production department and you will be notified of the publication date in due course.

With kind regards,

Anita Estes
